# Contextualizing biological perturbation experiments through language

**Menghua Wu**[*][†]
Massachusetts Institute of Technology
Cambridge, MA, USA

**Russell Littman, Jacob Levine**
Biology Research & AI Development, Genentech
South San Francisco, CA, USA

**Lin Qiu**[†]
Meta AI
Menlo Park, CA, USA

**David Richmond, Tommaso Biancalani, Jan-Christian Hütter**[*]
Biology Research & AI Development, Genentech
South San Francisco, CA, USA

## Abstract

High-content perturbation experiments allow scientists to probe biomolecular systems at unprecedented resolution, but experimental and analysis costs pose significant barriers to widespread adoption. Machine learning has the potential to guide efficient exploration of the perturbation space and extract novel insights from these data. However, current approaches neglect the semantic richness of the relevant biology, and their objectives are misaligned with downstream biological analyses. In this paper, we hypothesize that large language models (LLMs) present a natural medium for representing complex biological relationships and rationalizing experimental outcomes. We propose PERTURBQA, a benchmark for structured reasoning over perturbation experiments. Unlike current benchmarks that primarily interrogate existing knowledge, PERTURBQA is inspired by open problems in perturbation modeling: prediction of differential expression and change of direction for unseen perturbations, and gene set enrichment. We evaluate state-of-the-art machine learning and statistical approaches for modeling perturbations, as well as standard LLM reasoning strategies, and we find that current methods perform poorly on PERTURBQA. As a proof of feasibility, we introduce SUMMER (SUMMarize, retrievE, and answeR), a simple, domain-informed LLM framework that matches or exceeds the current state-of-the-art.[1]

## 1 Introduction

A fundamental paradigm for discovering causal relationships in molecular biology is intervention followed by measurement. Recent experimental methods like Perturb-seq allow biologists to manipulate the RNA and protein expression levels of each gene, and read out the effects on every other gene (Dixit et al., 2016; Datlinger et al., 2017; Replogle et al., 2022). While these experiments promise large-scale, unbiased insights, the measurement modality (single-cell sequencing) poses a significant cost burden and yields datasets of varying statistical power (Nadig et al., 2024). These challenges motivate *in-silico* approaches for predicting cellular responses to novel perturbations, and for automatically extracting high-level findings from perturbation data.

Current approaches for perturbation response prediction generalize to unseen perturbations by connecting them to perturbations that have been seen, often via knowledge graphs (Roohani et al., 2023). However, these approaches reduce textually-rich biological relationships to adjacency matrices, leading to loss of information. Furthermore, these methods are trained to regress the change in levels of genes upon perturbation: a task that is a precursor, but does not directly translate to downstream analyses like differential gene expression (Love et al., 2014) and gene set enrichment (Subramanian et al., 2005). Finally, most existing methods are black-box, revealing little about the learned biology without post-hoc probing.

---

[*]Correspondence to `rmwu{at}mit.edu` and `huetter.janchristian-klaus{at}gene.com`.
[†]Work completed while employed at Genentech.

[1]Our code and data are publicly available at `https://github.com/genentech/PerturbQA`

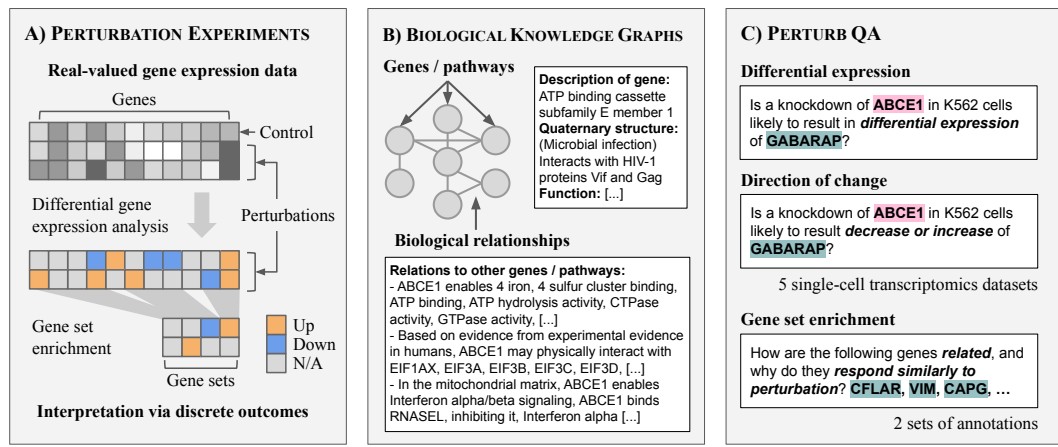

Figure 1: A) Perturb-seq experiments result in a matrix of gene expression levels, which are interpreted through discrete outcomes. B) Textually-rich, biological knowledge graphs can help explain these outcomes. C) Based on this premise, PERTURBQA introduces three tasks: predicting differential expression and direction of change for *unseen* perturbations, and summarizing data-driven gene clusters into cohesive sets.

We posit that language is a natural medium for traversing the structured, biological knowledge relevant to perturbation experiments. Based on this hypothesis, we propose PERTURBQA, a set of biological tasks that query discrete outcomes of perturbation experiments through question-answering. For example, rather than predicting the real-valued change in gene, we might ask, "does perturbation $p$ cause differential expression of $g$?" These tasks are inspired by the standard analysis pipeline for interpreting perturbation experiments, and compared to most biological reasoning benchmarks (Rein et al., 2023), they are *predictive* in nature. Ground truth labels are derived from five high quality single-cell RNA sequencing datasets with CRISPR interference (CRISPRi) perturbations (Replogle et al., 2022; Nadig et al., 2024), based on strict statistical considerations. Evaluation of state-of-the-art statistical, graph, and language-based methods reveal that these tasks are still far from solved.

To demonstrate that language-based reasoning can be effective on these tasks, we develop a simple LLM-based framework that matches or exceeds the current state-of-the-art on PERTURBQA. SUMMER (SUMMarize, retrievE, and answeR) is an inference-time strategy that incorporates standard LLM techniques alongside experimental data and biological knowledge graphs. An LLM is first asked to *summarize* textual descriptions associated with genes – as well as their impacts on and influences from other biological entities. In addition to "featurizing" genes, this step can be applied iteratively to characterize clusters of genes that exhibit similar responses or effects. Next, inspired by retrieval-augmented generation (Gao et al., 2023), we *retrieve* perturbation-gene pairs from existing experimental data based on knowledge graph proximity. While perturbation experiments are not textual in nature, their discretized outcomes can ground the LLM's reasoning and prevent hallucinations. Finally, inspired by chain-of-thought (Wei et al., 2022), the LLM *answers* the biological question via guided prompting, incorporating the knowledge graph summaries and retrieved experimental outcomes. To summarize, our contributions are three-fold.

1. We propose that biological perturbations should be modeled on the level of discrete outcomes that reflect downstream analyses, and that language models are suitable for capturing the relevant biology.

2. PERTURBQA is a set of real and currently unsolved tasks that evaluate how models reason over textually-rich, structured knowledge to discover new biology. We find that the current state-of-the-art performs poorly on this benchmark, which we hope will increase the accessibility and interest in machine learning modeling for biological perturbations.

3. We introduce SUMMER, a domain-informed LLM baseline, which matches or exceeds the state-of-the-art without any finetuning. SUMMER is implemented using a lightweight 8B model and operates natively in biologist-interpretable language.

## 2 RELATED WORK

**Predicting perturbation responses** Since experimental costs scale with the number of experimental contexts (cell lines) and perturbations, a number of works have been proposed to infer the post-intervention distribution of cells. Their goal is to generalize to unseen perturbations (Roohani et al., 2023; Bai et al., 2024; Märtens et al., 2024), or unseen contexts (Bunne et al., 2023; Lotfollahi et al., 2019). This paper focuses on the former setting, as we aim to optimize, not replace, experiments. An orthogonal direction is to predict the combined effects of multiple perturbations (Roohani et al., 2023; Gaudelet et al., 2024; Lotfollahi et al., 2023). While these models could be particularly helpful for alleviating experimental burden and rationally designing drug combinations, there are limited evaluation data for combinatorial perturbations ($< 150$ pairs in Norman et al. (2019)). Since our goal is to create a trustworthy benchmark for perturbation modeling, we choose to focus on single gene perturbations, and leave this as an opportunity for when better datasets are available.

**Language modeling and biology** LLMs have been applied to many biology-adjacent tasks, with several relevant directions included here. Biological question-answering (Hendrycks et al., 2021; Hao et al., 2024a) and scientific coding (Laurent et al., 2024; Hou & Ji, 2023) are common benchmarks to assess LLM reasoning, but these works primarily focus on tasks that human specialists are already able to do. Hsu et al. (2024) uses LLMs to predict Gene Ontology terms (Ashburner et al., 2000) associated with known gene sets. In contrast, the focus of our gene set enrichment task is to characterize *data-driven* gene clusters, which may not be significantly enriched for any *known* gene set, but are of biological interest to understand. Finally, LLMs have been used towards active experimental design (Roohani et al., 2024; Qu et al., 2024).

More broadly, there are a number of single-cell (Rosen et al., 2024; Hao et al., 2024b; Kalfon et al., 2024; Cui et al., 2024) and biological sequence (Lin et al., 2023; Nguyen et al., 2024; Schiff et al., 2024) foundation models, trained over raw biological data (gene count matrices, protein/DNA sequences). In this paper, we approach biological knowledge through natural language, but multimodal integration of foundation models could be a promising future direction (Wang et al., 2024b).

**Language-based reasoning and structured knowledge** Significant research efforts have focused on improving LLM reasoning and reliability. Chain of thought (Wei et al., 2022) first demonstrated that explicitly instructing LLMs to reason enables them to solve much more complex tasks, compared to directly asking for answers. Subsequent works have explored how to navigate the space of "thoughts," or in-progress generations (Yao et al., 2024; Zhang et al., 2024). Orthogonally, retrieval augmented generation (RAG) Lewis et al. (2020) was developed to increase LLM reliability. Relevant documents are first identified from a corpus, typically based on an embedding index, to act as source material for reasoning. Instead of querying an index, Graph RAG (Edge et al., 2024) summarizes corpuses into hierarchical graph structures, for richer language-based comparisons. While these methods have seen significant success in natural language applications (Gao et al., 2023; Jiang et al., 2023), they are less straightforward to implement in biology, where the vast majority of papers are inaccessible behind paywalls, and only abstracts are available (Wang et al., 2024a). As a result, document retrieval-based methods are difficult to evaluate in this paper's setting. Our method primarily retrieves experimental outcomes (binary labels), rather than relevant literature.

Textually-rich knowledge graphs have been probed through language, both in the classical natural language processing literature (Guu et al., 2015) and in modern LLM literature (Jin et al., 2024; Besta et al., 2024). Language can provide embeddings to be processed by downstream graph models, or directly serve as the medium for prediction (Chen et al., 2024; Kau et al., 2024). In this work, we take the latter approach, which opens several design choices. Graphs can be embedded alongside text via parameter-efficient finetuning (He et al., 2024; Perozzi et al., 2024), directly serialized into language (Zhao et al., 2023), inform retrieval (Mavromatis & Karypis, 2024), or any combination of these options. To minimize the computational cost of our proof of concept, we serialize graph-based knowledge into text and use graph structure to inform retrieval.

## 3 BACKGROUND

**Modeling perturbations** A perturbation experiment can be represented by a matrix $X \in \mathbb{R}^{N \times D}$, where $N$ is the number of perturbations, $D$ is the number of measured entities, and entries $x_{p,g} \in X$ represent the change in levels of entity $g$ under perturbation $p$, relative to a control $p_0$ (Figure 1A).

For example, in a CRISPRi Perturb-seq experiment (Replogle et al., 2022), the level of gene $p$ is decreased, and the resultant change $x_{p,g}$ in gene $g$ is measured, for all genes $g \in \mathcal{G}$.

Roohani et al. (2023) (GEARS) first proposed the task of predicting *unseen* perturbation outcomes in Perturb-seq data. Given $X_{\text{train}} \subsetneq X$, whose rows correspond to perturbations $\mathcal{P}_{\text{train}}$, their goal was to complete the rows $X_{\text{test}}$, corresponding to unseen perturbations $\mathcal{P}_{\text{test}}$. To generalize to $\mathcal{P}_{\text{test}}$, GEARS and subsequent works leverage knowledge graphs that relate the two sets of perturbations (Figure 1B). Specifically, they address a node-level regression task, over the graph $G = (V, E)$, where $V$ is a set of biological entities (e.g., genes and pathways) and $E$ is a set of relationships.

There are several aspects of the prevailing formulation that diverge from the findings biologists derive from these experiments. First, when converting knowledge graphs into adjacency matrices, the semantics of each edge are discarded, as they are typically annotated in free text (Ashburner et al., 2000). This loss of information may negatively impact model performance, especially in finite data regimes, as biological knowledge graphs often contain hierarchical relationships of conflicting semantics. Second, a common objective (and metric) is the real-valued error between the predicted and true responses, computed over genes that actually do respond (Roohani et al., 2023; Bai et al., 2024). However, these genes are not known prior to actual experimentation, and their identity is of high biological interest. Log-fold change is also known to be noisy, and it can be inconsistent across biological replicates (Nadig et al., 2024). Finally, the goal of perturbation experiments is to understand the underlying biology, but current methods focus solely on recapitulating the data distribution, errors in which may propagate to downstream analyses. These considerations motivate the creation of PERTURBQA, which is centered around higher level outcomes, whose significance can be statistically quantified.

**Statistical conclusions** Biologists draw conclusions of the form "$p$ impacts gene or pathway $g$" through statistical techniques like differential expression (Love et al., 2014) and gene set (Subramanian et al., 2005) analyses. In differential expression analysis, one assumes that $x \sim P_x$, where $P_x$ is often taken to be approximately normal (Cui & Churchill, 2003) or negative binomial (Love et al., 2014; Ahlmann-Eltze & Huber, 2020). The goal is to test between

$$H_0 : x_{p,g} = 0 \qquad \text{and} \qquad H_1 : x_{p,g} \neq 0, \qquad (1)$$

where rejection of $H_0$ translates to "$g$ is differentially expressed under perturbation $p$ compared to the control perturbation $p_0$." Differentially expressed genes may also be assessed by their direction of change, i.e. $x_{p,g} \lesseqgtr 0$.

Due to biological and technical noise, the measurement of individual genes may be unreliable, motivating statistical analyses at the level of gene *sets*. A data-driven approach for identifying gene sets is to cluster the rows and/or columns of the expression matrix $X$ and test whether more members of well-characterized sets are present in these clusters than expected by chance (Huang et al., 2008). While these "enriched" gene sets serve as the basis for annotating data-driven clusters, they do not consider the context of each experiment, e.g., the profiled cell line. Furthermore, significance cutoffs are difficult to assess, as the inclusion or exclusion of genes in gene sets was determined manually. As a result, data-driven clusters may exhibit consistent behavior in the experiment but fail to be enriched for known biological phenomena, thus eluding annotation (Replogle et al., 2022).

## 4 CONTEXTUALIZING BIOLOGICAL PERTURBATIONS

Our hypothesis is that traversing biological knowledge through language not only enables us to predict perturbation effects, but also to rationalize perturbation outcomes. We develop PERTURBQA, a benchmark to assess structured reasoning over semantically-rich graphs, in the context of molecular biology (Section 4.1). These tasks are non-trivial, both for graph-based methods and naive large language model (LLM) applications (Section 6). To validate our hypothesis, we introduce SUMMER, a simple LLM-based approach that matches or exceeds the current state-of-the-art on PERTURBQA, by considering experimental outcomes in the context of domain knowledge (Section 4.2).

### 4.1 PERTURBQA

PERTURBQA is composed of three primary tasks evaluated over five real datasets (Figure 1C). These tasks reflect the experimental and computational workflow associated with perturbation experiments.

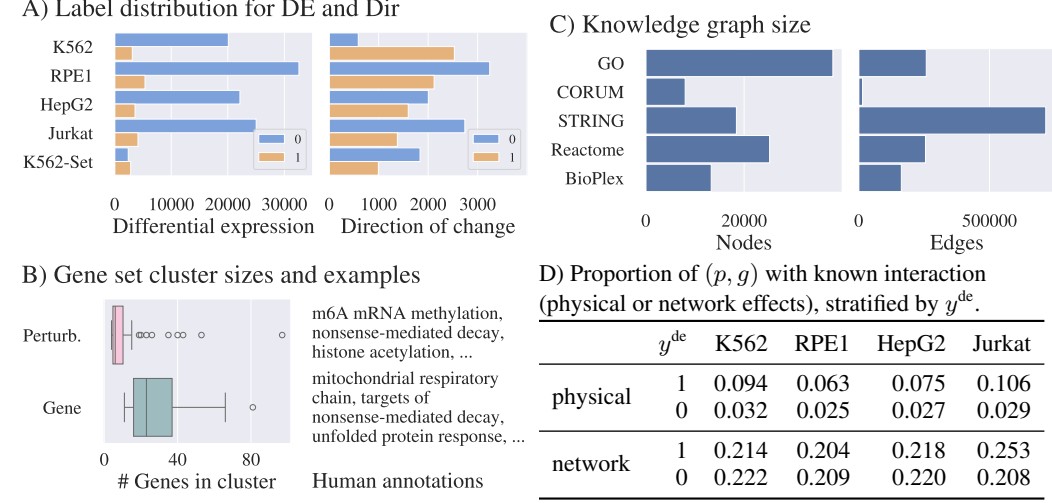

Figure 2: PERTURBQA dataset statistics. A) Differential expression and direction of change. B) Distribution of genes per cluster (gene set enrichment), with sample annotations. C) Knowledge graph sizes. D) DE genes are more likely to interact physically, but presence of interaction is minimally predictive (Table 1). There is little difference in network connectivity.

1. **Differential expression:** Given a pair of entities $(p, g)$, the goal is to predict $y_{p,g}^{\text{de}} \in \{0, 1\}$, where 0 indicates that perturbing $p$ results in no change to $g$, and 1 indicates that perturbing $p$ results in differential expression of $g$.

2. **Direction of change:** Given a pair of entities $(p, g)$, the goal is to predict $y_{p,g}^{\text{dir}} \in \{0, 1\}$, where 0 indicates that levels of $g$ decrease under perturbation $p$, and 1 indicates that levels of $g$ increase. This task is only evaluated on pairs for which $y^{\text{de}} = 1$.

3. **Gene set enrichment:** Given a set of genes $\mathcal{P}$, the goal is to identify a description $s$ that characterizes why members of $\mathcal{P}$ exert a consistent effect when perturbed ("perturbation cluster"), or respond similarly to perturbation ("gene cluster"). As ground truth, we leverage expert gene set annotations, published by the authors of Replogle et al. (2022).

**Perturb-seq datasets** We constructed our benchmark based on five Perturb-seq datasets, derived from Replogle et al. (2022) and Nadig et al. (2024). For each dataset, we identified differentially-expressed genes (DEGs) per perturbation using the Wilcoxon signed-rank test (Wilcoxon, 1945), resulting in pairs $(p, g)$ with associated labels $y_{p,g}^{\text{de}}$ and $y_{p,g}^{\text{dir}}$. Datasets are split 75:25 into train and test along the perturbation axis, with similar distributions of number of DEGs. To ensure label quality, we set a rigorous cut-off for DEGs and non-DEGs based on consistency across biological replicates and/or adjusted p-value (details and statistical analyses in Appendix A.2). The label distribution on the test set is depicted in Figure 2A.

Differential expression and direction of change are assessed at the granularity of single genes (K562, RPE1, HepG2, Jurkat) and gene sets (K562-Set), where the gene set is represented as single entities, with the mean expression of their constituents. Gene set enrichment is evaluated over K562-Set, where human annotations are taken as the ground truth. Figure 2B illustrates the distribution of the cluster sizes and example annotations.

**Domain knowledge** PERTURBQA tests whether models can effectively leverage structured domain knowledge and contextual information. Thus, in addition to test examples, we provide:

1. Harmonized and parsed knowledge graphs, with identifiers aligned to the perturbation data (Figure 2C). These provide high-quality, biological insights to aid reasoning.

2. Train examples (observation outcomes), to be used as a retrieval corpus or for model training. These may be useful for conditioning the predictions on each dataset, as perturbation responses may differ by cell line (Nadig et al., 2024).

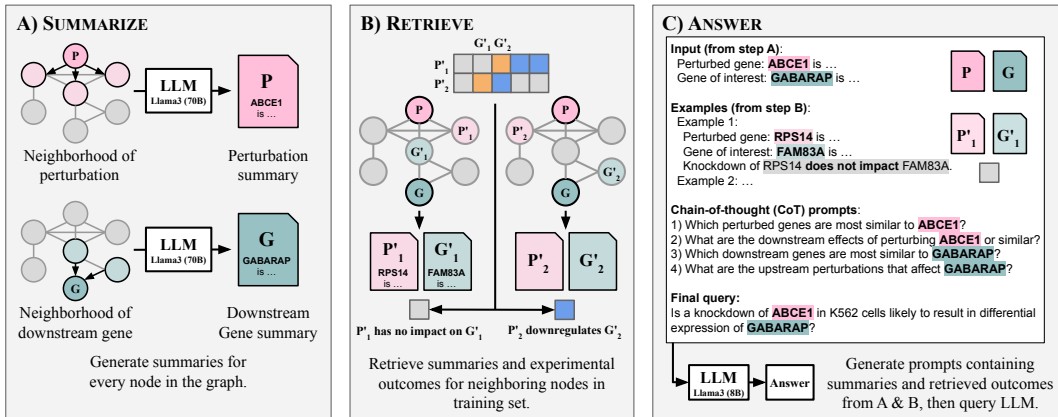

Figure 3: Overview of SUMMER. A) Knowledge graph entries are summarized per gene as both a perturbation $p$ and as a downstream gene $g$. B) Given a new pair $(p, g)$, sample related pairs $(p', g')$ with associated experimental outcomes. C) Concatenate summaries, examples, and guiding questions as prompt for LLM. Depicted prompt edited for concision. Full prompts in Appendix C.

In contrast to scientific literature, knowledge graphs densely distill key findings, either through human curation, or from individual (i.e. internally consistent) large-scale experiments. To ensure coverage of poorly characterized genes, we took the union of multiple sources, maintaining attribution. These include UniProt (Consortium, 2022), Ensembl (Martin et al., 2023), Gene Ontology (Ashburner et al., 2000; Aleksander et al., 2023), CORUM (Tsitsiridis et al., 2022), STRING (Szklarczyk et al., 2022), Reactome (Milacic et al., 2023), and BioPlex (Huttlin et al., 2021).

**A benchmark for discovery** PERTURBQA draws upon experimental assays and knowledge graphs to "connect the dots" between known biology and unanswered questions. A key concern is whether these questions are indeed unanswered, within the broader literature and by extension, current LLM weights. To quantify this, we compare the connectivity of pairs $(p, q)$, stratified by the differential expression label (Figure 2D). Only ∼3% of gene pairs in our test sets physically interact in *any* context, including other animals, and only ∼20% share any annotation, including at the coarsest levels. There is little difference between positive and negative pairs in terms of higher-level connectivity. Physically interacting genes are more likely to result in differential expression in our dataset, but presence of a physical interaction is minimally predictive of differential expression (Table 1). Finally, Nadig et al. (2024) was published strictly after we downloaded the knowledge graphs (June 2024). While the cell lines in question have been studied in other contexts, Nadig et al. (2024) released the first large-scale Perturb-seq screens in these two cell lines. Therefore, we conclude that these are indeed predictive tasks, rather than knowledge recall.

## 4.2 OVERVIEW OF SUMMER

SUMMER is an inference-time framework that consists of three steps, which draw inspiration from different aspects of message-passing neural networks and large language model reasoning strategies (Figure 3). First, we "featurize" each gene by summarizing database descriptions of their known function, and their relationships to other entities. To capture directionality, we generate two summaries for each gene: as a perturbation, and as a downstream gene. Next, we use the "training" set of seen perturbations as a retrieval corpus, where relevant perturbation-gene pairs are selected based on knowledge graph proximity. These pairs contribute both gene summaries and discretized experimental outcomes. Finally, we formulate a set of questions to guide the LLM's reasoning via chain-of-thought style prompting. Full prompts may be found in Appendix C.

**Neighborhood aggregation as summarization** Inspired by message-passing on graphs (Kipf & Welling, 2017), we characterize genes and their relationships to other biological entities by *summarizing* their graph neighborhoods. Given a gene $v$, we convert to natural language: knowledge graph entries $\{t_v\}$, describing node $v$, and $\{t_{v \to v'}\}$, describing its relationships with neighbors $v'$. As illustrated in Figure 3A, we concatenate all entries $t$ to generate two versions of each gene summary $s_v$. One asks for the downstream pathways that may be affected by the gene ($v$ as perturbation $p$), and the other focuses on the upstream pathways that may affect the gene ($v$ as downstream gene $g$).

We can apply this step hierarchically to characterize gene sets. Specifically, to summarize cluster $\mathcal{P}$, we concatenate single gene summaries $[s_v]_{v \in \mathcal{P}}$ as input to the LLM, with an emphasis on downstream/upstream effects, as appropriate. A variant of this is used to address the gene-set enrichment task, asking the LLM for both a short description and a name for the gene set (example in D.1).

**Graph-based retrieval of relevant training samples** Let $\mathcal{N}(v)$ represent the top $k = 10$ nodes $v'$ that share the most neighbors with $v$ in $G$. For each pair $(p, g)$, we sample up to 15 examples $(p', g')$ from $X_{\text{train}}$ that match the following criteria.[2]

1. Up to 5 pairs where both the perturbation and the downstream gene are related to $p$ and $g$ respectively, i.e. $p' \in \mathcal{N}(p)$ and $g' \in \mathcal{N}(g)$.
2. Up to 5 pairs with any $g'$ and where the perturbation is related to $p$, i.e. $p' \in \mathcal{N}(p)$.
3. Up to 5 pairs with any $p'$ and where the downstream gene is related to $g$, i.e. $g' \in \mathcal{N}(g)$.

These pairs are inserted into the prompt through their respective summaries $(s_{p'}, s_{g'})$ and the discretized experimental outcome $y_{p',g'}$ (Figure 3B).

**Perturbation outcome prediction as question-answering** To avoid hallucinations, we found it necessary to dictate that the LLM should consider both the textual summaries and experimental outcomes. Specifically, for both $p$ and $g$, we ask the LLM to identify the most similar $p'$ and $g'$ among those sampled, and to summarize their observed effects. For example, the prompt notes that the LLM should consider compensatory mechanisms, in addition to upstream or downstream pathways. Finally, we ask the LLM to answer the overall question, while citing specific retrieved examples. Figure 3C depicts a sketch of the process (example in D.2).

**Implementation details** We ran all experiments with Llama3 (Dubey et al., 2024) with default parameters of top $p$ 0.9 and temperature 0.6, using the LMDeploy framework (Contributors, 2023). Due to computational limitations, the gene summaries were generated by the 70B model, while all other inference utilized the 8B model. To quantify uncertainty, we ran the retrieval step three times and report the average prediction over these three inference runs.

## 5 EXPERIMENTAL SETUP

### 5.1 BASELINES

**Differential expression and direction of change** We benchmark a variety of baselines for unseen perturbation response prediction. All baselines were run with their published code and best reported hyperparameters, where applicable. **PHYSICAL** is a naive baseline that predicts 1 if $(p, q)$ are known to physically interact in any animal (Figure 2D, STRINGDB (Szklarczyk et al., 2022)) and 0 otherwise (DE only). **GAT** (Veličković et al., 2017) is a graph attention network trained with a ternary (up, down, no change) classification objective over the same knowledge graphs used to generate the prompts for SUMMER. This setup quantifies the information content contained in graph adjacencies alone. **GEARS** (Roohani et al., 2023) is a state-of-the-art graph attention network trained over the Gene Ontology (Ashburner et al., 2000) and gene co-expression graphs with a *regression* objective that focuses on differentially expresses genes. We use absolute predicted log-fold change for differential expression, and signed log-fold change for direction of change. **SCGPT** (Cui et al., 2024) is a Transformer-based, single-cell foundation model, which is finetuned for perturbation effect prediction with the GEARS regression objective.

We also consider language-centric baselines and ablations for SUMMER. **GENEPT** (Chen & Zou, 2024) encodes textual descriptions of genes (-GENE) and their protein (-PROT) products using commercial OpenAI embedding models, trained on natural language. These embeddings are input to a logistic regression classifier, trained separately for differential expression and direction of change. Recent work has reported that this baseline achieves the state-of-the-art on the regression formulation (Märtens et al., 2024). **LLM (No CoT)** provides the LLM with two examples (one of each label) and directly asks for the final answer without explanation. **LLM (No retrieval)** emulates our chain-of-thought style questioning, but does not retrieve any experimental outcomes. Instead, we

---

[2]The number of examples was chosen heuristically, so that all input prompts and potential outputs fit within the Llama3 8k token context window.

provide the LLM with a hypothesis (each answer option is sampled twice) and ask the LLM to extract supporting and refuting evidence from the gene summaries, before answering the question. Finally, to understand the information content in our retrieved samples, **Retrieval (No LLM)** takes the mean label over $(p', g')$ without appealing to the LLM for further processing.

**Gene set enrichment** We compare to gene set over-expression analysis (Fang et al., 2022), run over the gene clusters with a variety of gene set libraries – Gene Ontology, Reactome, CORUM – as well as their union (Combined). We take the concatenation of the top $k$ gene set names as the predicted summary, where gene sets are ordered by the size of their intersection with each cluster.

## 5.2 Metrics

**Differential expression and direction of change** It has been reported that gene responses tend to be correlated across perturbations (Kernfeld et al., 2023), e.g., stress response genes respond promiscuously. In addition, methods like GEARS and SCGPT predict real-valued change in genes, which yield rankings rather than strict probabilities. Thus, we compute binary AUROC over the predictions associated with each downstream gene, and take the average over downstream genes, corresponding to a macro AUROC score over downstream genes.

**Gene set enrichment** We consider both automated and human evaluation. Our ground truth consists of short textual descriptions (under 10 words), while our predictions and baseline outputs are long and vary in style. Standard text generation metrics like BLEU (Papineni et al., 2002) or ROUGE F1 (Lin, 2004) do not account for this difference in length as they were designed for machine translation. To assess whether the predictions adequately cover the ground truth annotation without penalizing for longer lengths, we report ROUGE-1 recall and BERT Score (Zhang et al., 2020), using BioBERT 1.2 (Lee et al., 2020), which was finetuned on 1M biological texts (18B words).

Due to the open-ended nature of the gene set task, automated evaluation methods are limited in their ability to reflect practical utility. Since this paper focuses on providing value to biologists, we recruited a domain specialist (molecular biologist, *not* the original annotator). We asked them to decide whether the top gene sets or LLM summaries were more informative, and whether the LLM summaries captured the same biology as the manual annotation (Section B.1). For future works, we share all LLM summaries in the data distribution for independent evaluation. If access to human experts is challenging, we also encourage LLM assessment of these questions.

## 6 Results

### 6.1 Differential expression and direction of change

We evaluated a number of state-of-the-art baselines on the differential expression and direction of change tasks (Table 1), and the results indicate that PERTURBQA tasks are largely unsolved. GEARS and SCGPT performances are close to random on differential expression, often exceeded by the naive PHYSICAL baseline. This may be due to the focus on change in differentially-expressed genes in their objective, instead of distinguishing between DEGs and non-differentially expressed genes. On other hand, GEARS is decent at direction of change in 3 of 5 cases, reflecting that its directionality loss may be more effective here.

GENEPT is a strong baseline, demonstrating the benefits of textual information towards these tasks. In terms of language-based reasoning, however, we observe that LLM (No CoT) and LLM (No retrieval) both perform no better than random guessing – highlighting that retrieving experimental outcomes and guiding LLM reasoning are both essential to completing this task. This is also reflected in the strong performance of Retrieval (No LLM). SUMMER is able to extract more value than "sum of its parts" in 7 of 10 cases, achieving the highest AUC in 8 of 10 cases.

Compared to methods that exclusively model knowledge graph connectivity, LLM outputs are directly interpretable by domain experts (Appendix D), to understand the model's shortcomings and provide context for the observed experimental outcomes. We studied 300 generations (3 trials of 100 DE examples) to identify primary failure modes (Appendix B.3). Incorrect causal directionality is a common error. For example, if A is related to C, which is *upstream* of B, A should *not* be affected when we perturb B. However, the LLM is unaware that C is upstream of B, so it predicts that A

Table 1: Results on differential expression and direction of change as binary prediction. AUROC is computed over the predictions associated with each gene, and averaged over perturbations. Standard deviation is reported over 3 runs (where applicable) or 3 rounds of sub-sampling. For more details, see Appendix A.4.

| Task | Model | K562 | RPE1 | HepG2 | Jurkat | K562-Set |
|---|---|---|---|---|---|---|
| | PHYSICAL | 0.53 | 0.52 | 0.52 | 0.54 | 0.55 |
| | GAT | $0.55_{\pm.02}$ | $0.57_{\pm.02}$ | $0.57_{\pm.02}$ | $0.55_{\pm.03}$ | $0.54_{\pm.01}$ |
| | GEARS | $0.54_{\pm.01}$ | $0.50_{\pm.01}$ | $0.48_{\pm.02}$ | $0.51_{\pm.01}$ | $0.49_{\pm.01}$ |
| | SCGPT | $0.52_{\pm.00}$ | $0.52_{\pm.00}$ | $0.48_{\pm.00}$ | $0.51_{\pm.00}$ | $0.52_{\pm.00}$ |
| Differential expression | GENEPT-GENE | $0.57_{\pm.02}$ | $0.54_{\pm.00}$ | $0.55_{\pm.02}$ | $0.55_{\pm.01}$ | $0.58_{\pm.01}$ |
| | GENEPT-PROT | $0.57_{\pm.01}$ | $0.56_{\pm.00}$ | $0.54_{\pm.01}$ | $0.55_{\pm.01}$ | $0.58_{\pm.01}$ |
| | LLM (No CoT) | $0.52_{\pm.01}$ | $0.51_{\pm.00}$ | $0.51_{\pm.01}$ | $0.52_{\pm.00}$ | $0.50_{\pm.00}$ |
| | LLM (No retrieval) | $0.51_{\pm.01}$ | $0.48_{\pm.00}$ | $0.49_{\pm.01}$ | $0.49_{\pm.01}$ | $0.50_{\pm.01}$ |
| | Retrieval (No LLM) | $0.58_{\pm.02}$ | $\mathbf{0.58}_{\pm.01}$ | $0.55_{\pm.00}$ | $0.55_{\pm.01}$ | $\mathbf{0.64}_{\pm.00}$ |
| | SUMMER | $\mathbf{0.60}_{\pm.00}$ | $\mathbf{0.58}_{\pm.00}$ | $\mathbf{0.61}_{\pm.00}$ | $\mathbf{0.58}_{\pm.00}$ | $0.61_{\pm.00}$ |
| | GAT | $0.58_{\pm.06}$ | $0.60_{\pm.04}$ | $0.64_{\pm.05}$ | $0.59_{\pm.04}$ | $0.53_{\pm.03}$ |
| | GEARS | $\mathbf{0.64}_{\pm.01}$ | $0.60_{\pm.01}$ | $0.52_{\pm.01}$ | $0.51_{\pm.01}$ | $0.59_{\pm.02}$ |
| | SCGPT | $0.48_{\pm.00}$ | $0.53_{\pm.00}$ | $0.51_{\pm.00}$ | $0.51_{\pm.00}$ | $0.54_{\pm.00}$ |
| Direction of change | GENEPT-GENE | $0.53_{\pm.05}$ | $0.57_{\pm.03}$ | $0.58_{\pm.03}$ | $0.57_{\pm.02}$ | $0.56_{\pm.02}$ |
| | GENEPT-PROT | $0.57_{\pm.01}$ | $0.57_{\pm.02}$ | $0.55_{\pm.01}$ | $0.58_{\pm.03}$ | $0.57_{\pm.02}$ |
| | LLM (No CoT) | $0.50_{\pm.01}$ | $0.49_{\pm.00}$ | $0.49_{\pm.00}$ | $0.50_{\pm.01}$ | $0.50_{\pm.01}$ |
| | LLM (No retrieval) | $0.49_{\pm.04}$ | $0.52_{\pm.03}$ | $0.51_{\pm.06}$ | $0.53_{\pm.05}$ | $0.45_{\pm.18}$ |
| | Retrieval (No LLM) | $0.50_{\pm.00}$ | $0.50_{\pm.00}$ | $0.50_{\pm.00}$ | $0.50_{\pm.00}$ | $0.50_{\pm.00}$ |
| | SUMMER | $0.62_{\pm.01}$ | $\mathbf{0.64}_{\pm.01}$ | $\mathbf{0.65}_{\pm.00}$ | $\mathbf{0.66}_{\pm.01}$ | $\mathbf{0.69}_{\pm.01}$ |

Table 2: Gene set enrichment on K562 genome-wide clusters. Metrics reported are ROUGE-1 recall, as well as BERT Score precision, recall, and F1, computed with BioBERT-1.2. Since the baselines are statistical methods, they are not subject to stochasticity.

| | | Gene clusters | | | | Perturbation clusters | | | |
|---|---|---|---|---|---|---|---|---|---|
| Enrichment | Top | $R_{\text{ROUGE1}}\uparrow$ | $P_{\text{BERT}}\uparrow$ | $R_{\text{BERT}}\uparrow$ | $F_{\text{BERT}}\uparrow$ | $R_{\text{ROUGE1}}\uparrow$ | $P_{\text{BERT}}\uparrow$ | $R_{\text{BERT}}\uparrow$ | $F_{\text{BERT}}\uparrow$ |
| Gene Ontology | 5 | 0.17 | 0.64 | 0.66 | 0.62 | 0.38 | 0.66 | 0.72 | 0.68 |
| Gene Ontology | 10 | 0.32 | 0.60 | 0.65 | 0.60 | 0.60 | 0.62 | 0.71 | 0.65 |
| Reactome | 5 | 0.18 | 0.60 | 0.65 | 0.60 | 0.49 | 0.60 | 0.68 | 0.62 |
| Reactome | 10 | 0.27 | 0.54 | 0.64 | 0.56 | 0.59 | 0.56 | 0.67 | 0.60 |
| CORUM | 5 | 0.07 | 0.63 | 0.45 | 0.42 | 0.45 | 0.64 | 0.63 | 0.60 |
| CORUM | 10 | 0.07 | 0.61 | 0.44 | 0.41 | 0.47 | 0.61 | 0.62 | 0.58 |
| Combined | 5 | 0.14 | 0.62 | 0.65 | 0.61 | 0.41 | 0.63 | 0.71 | 0.66 |
| Combined | 10 | 0.27 | 0.59 | 0.65 | 0.59 | $\mathbf{0.63}$ | 0.57 | 0.69 | 0.62 |
| SUMMER (8b) | desc | $\mathbf{0.57}$ | 0.63 | 0.76 | 0.69 | 0.26 | 0.63 | 0.75 | 0.68 |
| SUMMER (8b) | name | 0.20 | $\mathbf{0.74}$ | 0.76 | $\mathbf{0.75}$ | 0.12 | 0.75 | 0.76 | 0.75 |
| SUMMER (70b) | desc | 0.45 | 0.63 | $\mathbf{0.77}$ | 0.69 | 0.59 | 0.65 | 0.80 | 0.72 |
| SUMMER (70b) | name | 0.15 | 0.73 | 0.76 | 0.74 | 0.37 | $\mathbf{0.77}$ | $\mathbf{0.82}$ | $\mathbf{0.79}$ |

changes when B is changed. This suggests that additional domain knowledge or more structured reasoning strategies may be necessary.

## 6.2 GENE SET ENRICHMENT

SUMMER is able to recapitulate human cluster annotations, exceeding over-representation analysis across automated metrics (Table 2). In particular, SUMMER performs much better than the baselines on gene clusters, which are larger than the perturbation clusters (Figure 2B) and more enriched for generic gene sets (Table 3, row 1). Based on independent human evaluation, the LLM-generated summary was similar or more informative than the classical gene set enrichment results in 92% of cases, and agrees with the independent annotator in 72% of cases (Appendix B.1). In the 28% that disagreed with the independent annotator, many clusters contain niche information, not found

Table 3: Gene set enrichment on K562, example outputs. Left: Human annotation. Middle: Top over-represented gene sets (combined). Right: Proposed gene set name in bold, followed by description. Row 1 is a gene cluster. Rows 2-3 are perturbation clusters.

| Genes | Annotation | Gene sets (combined) | SUMMER (8b) |
|---|---|---|---|
| SLC25A5, EIF4B, SLC25A3, RPL3, . . . , RPL41 | translation | cytosol, RNA binding, cytoplasm, membrane, GTP hydrolysis | **Ribosomal and Translation Regulation Gene Set** The gene set is comprised of genes involved in protein synthesis, translation, and ribosomal function, with a focus on the regulation of protein synthesis and the assembly of ribosomal subunits. |
| ZC3H13, CBLL1, METTL14, METTL3, PSMG1, RBM15 | m6A mRNA methylation | nucleus, nucleo plasm, RNA N6-methyladenosine methyltransferase complex, mRNA processing | **M6A Methylation Complex-Associated Genes.** The gene set is composed of genes involved in the regulation of N6-methyladenosine (m6A) methylation of RNAs, influencing mRNA splicing and RNA processing. These genes are associated with the WMM complex and interact with each other to modulate gene expression. |
| CMTR2, RBM14-RBM4, RBM4, UNCX, WDFY3 | unknown | no significant sets | **RNA Processing and Regulation Gene Set.** The gene set is composed of genes involved in RNA processing and regulation, including mRNA cap modification, alternative splicing, and RNA-binding activities. These genes converge on pathways related to mRNA stability, translation, and cellular differentiation. |

in typical databases (Table 7). We also observe that in difficult cases, gene set over-representation analysis tends to focus on highly specific gene sets, which cover subsets of these clusters. The LLM takes the opposite approach, and its summaries tend to "lift" the description to higher levels of hierarchy (Table 8). While the two strategies provide orthogonal information, the LLM's outputs are more coherent. Finally, SUMMER also characterizes clusters for which no gene sets were enriched, and thus could not be annotated manually (Table 3, row 3). These clusters tend to be smaller, or exhibit lower agreement.

## 7    CONCLUSION

In this work, we proposed PERTURBQA, a benchmark for language-based reasoning over structured data that arise from real biological problems. We evaluated a variety of state-of-the-art methods and showed that while these problems are feasible, they are far from solved. To address these tasks, we also introduced SUMMER, a LLM-based framework that draws upon both biological knowledge graphs and existing experimental data. SUMMER outperforms baselines on PERTURBQA, but leaves ample room for future study. We hope that this work will lower the barrier of entry into computational modeling of biological perturbation experiments and enable richer, more interpretable methods for these applications.

## ETHICS STATEMENT

While this work focuses broadly on discovering causal relationships in molecular biology, the methods described do not involve the design of potentially harmful chemical agents or other biomolecules. Our work uses publicly available datasets, which were generated in *in-vitro* laboratory settings.

## REPRODUCIBILITY STATEMENT

All datasets and code can be found at our repository: `https://github.com/genentech/PerturbQA`. Our data processing pipeline is described in detail in Appendix A.2. Templates of all prompts used for LLM experiments can be found in the code distribution and in Appendix C.

ACKNOWLEDGMENTS

We would like to thank Sandra Melo-Carlos and Jack Kamm for an introduction to Perturb-seq and guidance on data analysis; Aviv Regev for inspiring the gene set enrichment task; Umesh Padia for evaluating and providing feedback on our model outputs; and Romain Lopez, Alexander Wu, Heming Yao, Patrick Skillman-Lawrence, Xiaotian Ma, Taro Makino, Martin Rohbeck for continual feedback on this project.

This work was funded by Genentech. M.W., R.L., J.L., L.Q., T.B., D.R., and J.-C. H. were employees of Genentech while working on this project. J.L., L.Q., T.B., D.R., and J.-C. H. have equity in Roche.

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

## A  EXPERIMENTAL DETAILS

### A.1  K562 GENE SETS

We downloaded K562 genome-wide gene sets from the supplementary data associated with Figure 4B of Replogle et al. (2022). These gene sets were obtained by "cluster[ing] genes into expression programs based on their co-regulation." We evaluated gene set enrichment over clusters that were manually annotated, though we provide predictions over all gene sets (regardless of annotation status).

For differential expression analysis, we took the average of the Log(TP10k+1) values over each gene set, for each cell, similar to a pathway activity score (Wagle et al., 2018).

### A.2  DATA PROCESSING

We normalized all gene counts to Log(TP10k+1) values (log-transformed UMI count per 10k), where the count $c_{ij}$ of gene $j$ in cell $i$ is mapped to

$$\log \left( \frac{c_{ij}}{\sum_j c_{ij}} \cdot 10,000 + 1 \right). \tag{2}$$

To determine differentially expressed genes (DEGs), we ran the Wilcoxon signed-rank test (Wilcoxon, 1945) with Benjamini-Hochberg correction (Benjamini & Hochberg, 2000) between non-targeting control (NTC) cells and perturbed cells, for each perturbation.

To focus on perturbations with phenotypic effect, we excluded all perturbations that resulted in $\leq 5$ DEGs ($p < 0.05$), with the exception of 100 negative control perturbations (0 DEGs), sampled uniformly at random per dataset.

We selected thresholds based on statistical consistency (Section A.5). For the two K562 datasets, we defined "DE" perturbation - gene pairs as those shared between the datasets at $p < 0.05$. Non differentially-expressed pairs were sampled from those that were *not* differentially-expressed in *either* dataset. Since the remaining datasets lacked biological replicates, we defined "DE" pairs as those with $p < 0.01$, and sampled "non-DE" pairs from those with $p > 0.1$. We selected the top 20 DEGs per perturbation ranked by adjusted p-value as "positives." We sampled 100 non-DEGs per perturbation as "negatives."

To ensure similar distributions between our training and testing splits, we sorted both selected perturbations and DEGs based on prevalence. We split perturbations 75:25 between training and testing. Validation data were sampled at random during training (10% of training). Further details regarding dataset and data split statistics may be found in Tables 4 and 5.

Table 4: Data statistics. K562* non-targeting control cells were subsampled due to the size of the genome-wide dataset. K562-es⋆ (essential) was only used to filter K562 DE and non-DE genes.

| Dataset | Cells | | Perturbations | | | Features |
| --- | --- | --- | --- | --- | --- | --- |
| | Control | Perturbed | Total | Train | Test | |
| K562* | 5,000 | 919,124 | 9851 | 1564 | 267 | 4136 |
| K562-es⋆ | 10,691 | 299,645 | 2049 | — | — | — |
| RPE1 | 11,485 | 236,164 | 2354 | 1596 | 406 | 4760 |
| HepG2 | 4,976 | 140,497 | 2393 | 1086 | 278 | 7435 |
| Jurkat | 12,013 | 250,943 | 2392 | 1227 | 313 | 6842 |
| K562-Set | 5,000 | 919,124 | 9851 | 1401 | 357 | 20 |

### A.3  LLM DETAILS

Due to the stochastic nature of LLM generations, we noticed that the LLM would occasionally abstain from selecting one of the intended labels, due to insufficient evidence for either. To account

Table 5: Differential gene expression data split statistics. Number of perturbation - gene (set) pairs.

| Dataset | Split | Total | non-DE | Differentially expressed Total | Up | Down |
|---------|-------|-------|--------|-------|-----|------|
| K562 | Train | 134,467 | 117,606 | 16,861 | 11,041 | 5,820 |
|  | Test | 23,212 | 20,093 | 3,119 | 2,530 | 589 |
| RPE1 | Train | 149,147 | 127,860 | 21,287 | 8,381 | 12,906 |
|  | Test | 37,942 | 32,577 | 5,365 | 2,121 | 3,244 |
| HepG2 | Train | 101,140 | 86,883 | 14,257 | 6,249 | 8,008 |
|  | Test | 25,749 | 22,146 | 3,603 | 1,599 | 2,004 |
| Jurkat | Train | 113,684 | 97,747 | 15,937 | 5,119 | 10,818 |
|  | Test | 29,138 | 25,017 | 4,121 | 1,379 | 2,742 |
| K562-Set | Train | 20,606 | 9,367 | 11,239 | 3,953 | 7,286 |
|  | Test | 5,235 | 2,403 | 2,832 | 995 | 1,837 |

Table 6: Abstain rate on differential expression (DE) and direction of change (Dir) across all datasets.

| Model | DE | Dir |
|-------|-----|-----|
| LLM (No Retrieval) | 0.02 | 0.36 |
| LLM (No CoT) | $3.3 \times 10^{-6}$ | 0 |
| SUMMER | $8.9 \times 10^{-4}$ | 0.03 |

for this, we intentionally added "insufficient information" as a third answer option. We ran inference on each input sample at least 3 times and took the mean predicted label, after removing all abstaining outputs.

A small fraction of inputs $(p, g)$ resulted in no predictions after this filtering, or were unable to be parsed by our rule-based parsing. The latter is due to the insufficient capacity of Llama3 8B (relatively small LLM) to follow instructions. Since these examples differed by model, we substituted the prediction with an uninformed baseline (the mean label of $g$ over the training set) for evaluation. The final abstain rate varied based on LLM prompting strategy (Table 6). LLM (No Retrieval) abstained nearly a third of the time on direction of change. In contrast, LLM (No CoT) only abstained a single time, over all datasets. The improved instruction following may be due to the concise nature of the expected output (only a single answer). Finally, SUMMER nearly always produced a prediction over 3 runs (e.g. 0.08% abstain on DE).

## A.4 BASELINES

For GAT, we grid searched over the number of layers (1, 2, 4, 8) and hidden dimension (64, 128, 256). We used FFN dimension 1024 (memory constraint), GELU activation, dropout of 0.1, weight decay 1e-6, learning rate 1e-4, and residual connections. We selected the top models based on validation performance (arbitrary 10% of train). In addition to node features, GAT also learned edge attributes, which indicated the source knowledge graph of each edge.

For K562-Set, we pooled the mean embedding of each gene set's genes before the prediction head in GAT. On GEARS and SCGPT, we used the mean predicted log-fold change over each gene set's genes (mirrors data pre-processing). A small number of genes (97 out of 11,234) did not map to GENEPT embeddings. We set the embeddings for these genes to the mean perturbation / gene embedding in their respective training sets.

For uncertainty quantification, we used the top 3 runs for GAT. For GENEPT, since logistic regression does not inherently introduce randomness (unless it fails to converge; it always converges here), we subsampled 80% of the training set for each of 3 runs. Since GEARS and SCGPT operate over

single cells, rather than pseudo-bulk estimates, we subsampled 80% of the single cells before taking the average for each of 3 evaluations.

## A.5 STATISTICAL ANALYSIS

We provide empirical analyses on the quality of our datasets and labels. Figure 4 shows that the Wilcoxon rank sum test is relatively well-calibrated on our data, though the test tends to be conservative, erring on the side of identifying fewer DEGs. Thus, we selected a relative higher p-value threshold for negative examples. Figures 5 and 6 illustrate that K562 gene clusters and top DEGs are consistent across near-biological replicates (two experiments in the same cell line, by the same lab). This motivates both the K562-set setting, as well as our selection of the top DEGs as positives.

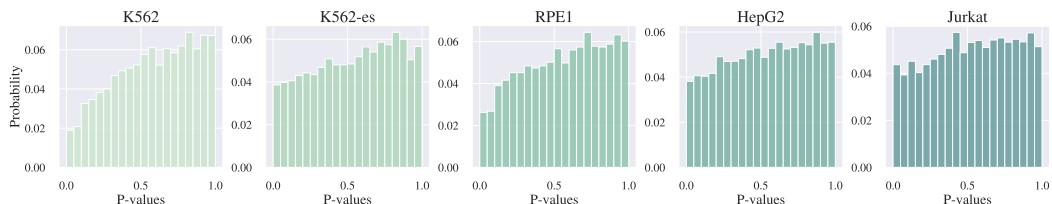

Figure 4: Assessing p-value calibration over single-cell datasets. We split the non-targeting controls (NTCs) randomly in half, and run the Wilcoxon test to compare the two halves. We would expect to see that the (non-adjusted) p-values are uniformly distributed between 0 and 1. Here, we see that the Wilcoxon test is slightly conservative, i.e. it leans towards reporting "non-differentially expressed."

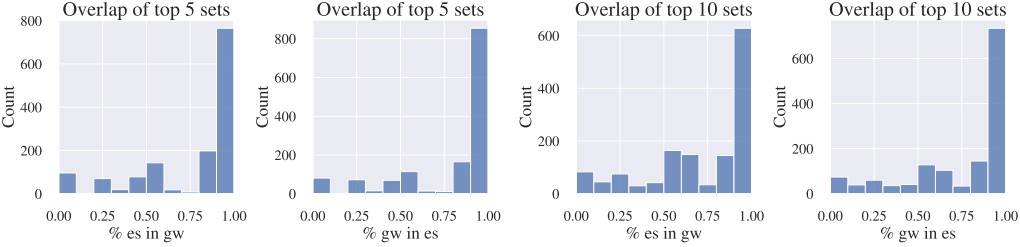

Figure 5: K562 gene clusters show consistent response between biological replicates. We compute the top $k = 5, 10$ significant gene clusters, sorted by adjusted p-value, for both K562 genome-wide and K562 essential. For each perturbation, we compute the percentage of shared gene clusters (normalizing by genome-wide and essential, respectively). We see that the clusters are relatively consistent across both datasets, with a high fraction of perfect overlaps.

## B  QUALITATIVE ANALYSIS

### B.1  HUMAN EVALUATION OF SUMMARIZATION RESULTS

Due to the open-ended nature of the gene set task, automated evaluation methods are limited in their ability to reflect practical utility. Since this paper focuses on providing value to biologists, we recruited a domain specialist (molecular biologist, trained in wet lab and computational biology) for this task. We presented them with a document formatted as follows and asked two questions.

```
1) Is A or B more informative, or about the same? Options: A, B, same
2) Does B capture the same biology as the bolded annotation? Options: yes, no
{ground truth label}: {list of genes}
A: {top 10 gene sets (all databases)}
B: {LLM-generated name}: {LLM-generated descriptions}
```

Overall, the LLM-generated summary is equal or better to the classical gene set enrichment results in 92% of cases, and agrees with the independent annotator in 72% of cases.

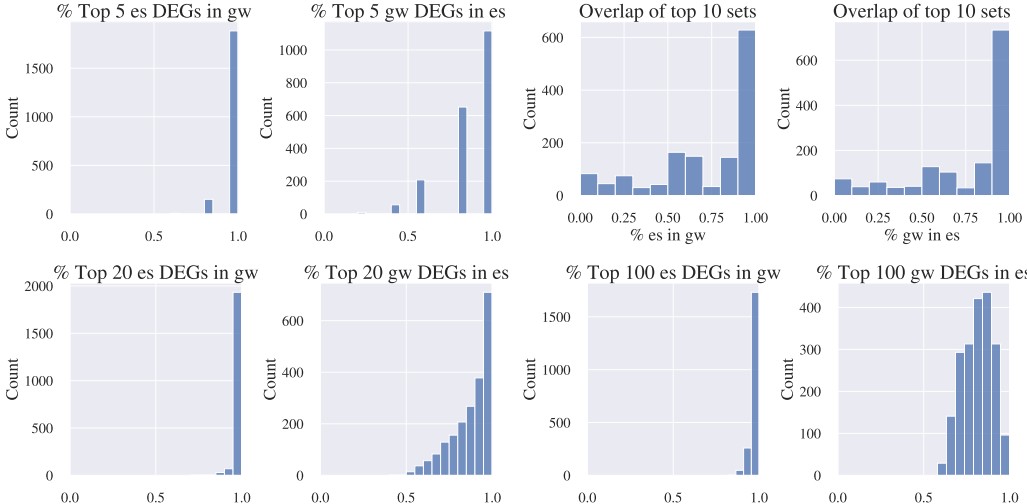

Figure 6: K562 DEGs are reasonably consistent at the top (ranked by p-value). For each perturbation, we plot the percentage of shared top = 5, 10, 20, 100 DEGs (normalizing by genome-wide and essential, respectively). In our final dataset, we took the intersection of the top 20 DEGs as "DE" and the intersection of the negatives as "non-DE."

1. In 21/25 cases, the biologist reported that the LLM-generated summary was more informative. In 2/25 cases, they contained the same amount of information; and in 2/25 cases, the gene set contained more information.

2. In 18/25 cases, the biologist reported that the LLM summary captured the same biology as the original human annotation (our ground truth labels).

We analyze the cases in which the LLM provides less information, or fails to capture the independent human annotation.

1. In the 2 cases where the gene sets contained *more information*, a list of specific protein complexes were discovered, e.g. below.

   ```
   Eukaryotic Translation Termination, EIF2AK4 (GCN2) dimer autophosphorylates, EIF2AK4 (GCN2)
   binds tRNA, Aminoacyl-tRNA binds to the ribosome at the A-site, 80S:Met-tRNAi:mRNA:SECISBP2:Sec-
   tRNA(Sec):EEFSEC:GTP is hydrolysed to 80S:Met-tRNAi:mRNA:SECISBP2:Sec and EEFSEC:GDP by
   EEFSEC, UPF1 binds an mRNP with a termination codon preceding an Exon Junction Complex, Transloca-
   tion of ribosome by 3 bases in the 3' direction, Translation of ROBO3.2 mRNA initiates NMD, Translation of
   ROBO3.2 mRNA is negatively regulated by NMD, The SRP receptor binds the SRP:nascent peptide:ribosome
   complex
   ```

   However, this output is difficult to parse, compared to the LLM-generated output, which faithfully and concisely represents the original annotation of "translation."

   ```
   Ribosomal Protein Components Involved in Translation: This gene set is comprised of components of the
   large and small ribosomal subunits, which are essential for protein synthesis and translation. These genes
   are involved in the assembly and function of the ribosome, facilitating the translation of messenger RNA into
   protein.
   ```

2. In the 7/25 cases where the LLM summary differed from the human annotation (Table 7), the LLM annotation tended to miss some highly specific terms, e.g. "targets of nonsense-mediated decay" was generalized to "stress response," and "dysregulated lncRNA antisense transcripts" was generalized to "nuclear gene regulation." Related terms tend to be sparsely annotated in Gene Ontology, so this indicates that it would be useful to tune the granularity of generations in the future, or to generate multiple candidates for specific descriptions.

## B.2 CAPTURING THE GREATEST COMMON DENOMINATOR

By construction, clusters of genes may vary in their degree of specificity and agreement. For example, "translation" contains over 20 million annotations in the Gene Ontology, while variants of

Table 7: Error analysis of gene set summaries that differed from the manual annotation, based on independent human evaluation. These summaries tend to be more generic than the original annotation.

| Annotation | SUMMER (8b) gene set name |
|---|---|
| targets of nonsense-mediated decay | RNA Processing and Stress Response Regulators |
| dysregulated lncRNA antisense transcripts | Mitochondrial & Nuclear Gene Regulation and Metabolism |
| unfolded protein response mTORC1 signaling | Amino Acid Metabolism and Protein Synthesis Regulators |
| NFkB signaling | Regulators of Cellular Homeostasis and Signaling |
| myeloid differentiation | Actin Cytoskeleton and Immune Response Regulators |
| growth signaling | Protein Trafficking and Transcriptional Regulation |
| growth targets of Myc | RNA Processing and Translation Regulation |

Table 8: Unannotated gene clusters are typically less specific and coherent. Compared to gene set enrichment, which tends to focus on highly-specific subsets of each clusters, SUMMER tends to identify the "greatest common denominator" among the genes.

| Genes | Gene sets (combined) | SUMMER (8b) |
|---|---|---|
| HCCS, AGPS, FAM136A, MTREX, PITHD1, KIF2A, REXO2, ..., EIF6 | 2Fe-2S is inserted in UQCRFS1; 39S ribosomal subunit; 3' incision by ERCC5 (XPG) in TC-NER; 55S ribosome; 1-palmitoylglycerone phosphate + hexadecanol => O-hexadecylglycerone phosphate + palmitate, ... | **Ribosomal and Mitochondrial Gene Set.** The gene set is composed of genes involved in various cellular processes, including protein synthesis, translation, and regulation of gene expression. Many of these genes are involved in the assembly and function of ribosomes, the translation of mitochondrial DNA-encoded genes, and the regulation of protein synthesis in response to cellular signals. |
| ATP6AP1, ITM2A, NPC2, CREG1, UQCRB, ..., HEXA | keratan sulfate catabolic process; transcription corepressor activity; transcription regulator complex; dermatan sulfate catabolic process, ... | **Protein Processing and Trafficking Gene Set.** The gene set is comprised of genes involved in various cellular processes, including protein transport, immune response, and cellular differentiation. Common pathways among these genes include protein processing and modification, as well as cellular trafficking and signaling. |
| RALA, STARD3NL, AKAP8L, ATP2B1, ..., YTHDF2 | Expression of Perilipin (PLIN); Dephosphorylation of inactive SRC by PTPB1; Expression of Phosphoenolpyruvate carboxykinase 1 (PEPCK-C), ... | **Regulatory Genes in Cellular Processes and Signaling.** The gene set is comprised of genes involved in various cellular processes, including protein synthesis and degradation, gene expression, transcription regulation, and cellular signaling. These genes are also involved in maintaining proper calcium homeostasis, regulating mitochondrial function, and facilitating protein trafficking and targeting. |

"nonsense-mediated decay" vary from 27 to 74,097 annotations (as of November 2024) (Ashburner et al., 2000).

Clusters that are smaller, or exhibit lower agreement, tend to elude manual annotation. In these cases, gene set over-representation analysis tends to focus on highly specific gene sets, which cover subsets of these clusters. The LLM takes the opposite approach, and its summaries tend to "lift" the description to higher levels of hierarchy (Table 8).

## B.3 ERROR AND INCONSISTENCY ANALYSIS

We analyzed 300 generations (3 trials of 100 DE examples) to understand the failure modes of SUMMER. Errors and inconsistencies primarily resulted from deductions backed by overly-generic information. For example, the LLM may list an excessively broad set of factors that may influence expression.

> Perturbations of genes involved in mitochondrial function, protein synthesis, or transcriptional regulation may potentially affect CC2D1A expression.

In several instances, the LLM may be confused between concepts which are loosely connected, but not in the same context.

> The genes of interest in the examples are QKI, MT-CYB, ADA, MT-ND1, and RPL12. Among these, MT-CYB and MT-ND1 are similar to TXNIP as they are both involved in mitochondrial function and electron transport, which is a related process to TXNIP's role in oxidative stress and mitochondrial function.

Here, "oxidative stress" *is* related to the mitochondria, but TXNIP is upstream of a number of stress response genes and is unrelated to healthy mitochondrial function. Thus, it should *not* respond similarly to MT-CYB. This aspect might be resolved with a higher-capacity model, but the 8b LLM cannot distinguish.

Finally, we observe that there is a fine line between abstaining and predicting "no effect" due to lack of information.

> Based on the analysis above, we can see that perturbing ABCE1 or similar genes did not result in differential expression of any genes, including CDK11A. Additionally, perturbing EIF3J resulted in differential expression of HSP90AA1, which is similar to CDK11A. However, the relationship between ABCE1 and CDK11A is not well-represented in these examples. Therefore, it is unlikely that a CRISPRi knockdown of ABCE1 in K562 cells will result in differential expression of CDK11A.
>
> Final answer: There is insufficient evidence to determine how knockdown of ABCE1 affects CDK11A.

## C  PROMPTS

We share the prompts used by each stage of SUMMER, as well as the LLM baselines. All text surrounded by {…} are replaced with example-specific inputs (Appendix D).

### C.1  SUMMARIZATION TEMPLATES

We generate gene summaries at two levels: single node and 1-hop knowledge graph neighborhoods. For both levels, we use two prompts per gene (as a perturbation, and as a downstream gene). Wherever gene sets are the downstream entities, we replace "gene" with "gene set."

The following is an example of a single-node summary of a perturbed gene.

> You are an expert molecular biologist who studies how genes are related using Perturb-seq.
>
> Task: You are writing a brief overview of the human gene {gene}, with a focus on its molecular and cellular functions. You will be provided a set of database entries about the gene. Ensure that your overview remains faithful to this domain knowledge.
>
> Format:
> - Write one to two sentences describing the primary molecular and cellular function of gene {gene}.
> - Write one sentence describing the potential downstream impact of perturbing gene {gene} via gene knockdown.
> Constraints:
> - Maintain a professional tone throughout.
> - Do not comment on your own writing.
> - Do not add any notes or references. Do not make up additional information.
> - Do not discuss the importance or impact of the gene. Focus only on its function.
> Domain knowledge: {description}
> Brief overview of gene {gene}:

To generate downstream gene descriptions, we replace the second formatting instruction with the following.

> Write one sentence describing what types of perturbations might impact the expression of gene {gene}. For example, you might consider pathways that are upstream of the gene or compensatory mechanisms.

Single-node summaries are input alongside additional knowledge graph relationships for 1-hop neighborhood summaries.

> You are an expert molecular biologist who studies how genes are related using Perturb-seq.
>
> Task: You are writing a brief overview of the human gene {gene}, with a focus on the downstream effects of perturbing {gene} via gene knockdown (loss of function).
> Inputs: You are provided
> - Description of perturbed gene {gene}

```
- Database entries relating {gene} to other genes or pathways
Format: Write up to five sentences describing the molecular and cellular impact of perturbing gene {gene} via gene
knockdown.
Constraints:
- Remain faithful to all domain knowledge. Do not make up additional information.
- Summarize all common aspects succinctly, but point out notable differences within these sets of genes.
- Maintain a professional tone throughout. Do not comment on your own writing. Do not add any notes or references.
- Omit the importance or impact of the gene. Focus only on its function.
- Omit all non-specific information and obvious statements, e.g. "this gene is involved in cellular processes."
Description of gene {gene}: {single-node summary}
Relations to other genes: {relationships}
Downstream effects of perturbing {gene} via gene knockdown:
```

To generate downstream gene descriptions, we replace the "Task" instruction with the following.

```
Task: You are writing a brief overview of the human gene {gene}, with a focus on molecular and cellular perturbations
that may affect the levels of gene {gene}. For example, you might consider pathways that are upstream of the gene or
compensatory mechanisms.
```

To generate gene set summaries (for differential expression and direction of change), we used the following prompt, where the manual annotation is provided as input.

```
You are an expert molecular biologist who studies how genes and pathways are related using Perturb-seq.
Task: You are writing a brief overview of a gene set associated with {manual annotation} in human cells. You will be
provided descriptions of the constituent genes. Ensure that your overview remains faithful to this domain knowledge.
Format:
- Write one to two sentences summarizing how the given genes are related. Be sure to characterize how they are associated
with {manual annotation}.
- Write one to two sentences describing what types of perturbations might impact the expression of the genes in this
gene set. For example, you might consider pathways that are upstream of these genes or compensatory mechanisms.
Constraints:
- Maintain a professional tone throughout.
- Do not comment on your own writing.
- Do not add any notes or references. Do not make up additional information.
- Do not discuss the importance or impact of the gene set. Focus only on its function.
Descriptions of the constituent genes: {list of gene summaries}
Brief overview of gene set ({manual annotation}):
```

Finally, for the gene set enrichment task, we use the following prompt for hierarchical summarization of gene sets. Note that the manual annotations are *not* provided here. We parse the outputs of "Brief overview of gene set" and "Name of gene set" as the description and name in our evaluation.

```
[Start of Prompt]
You are an expert molecular biologist who studies how genes and pathways are related using Perturb-seq.
Task: You are writing a brief overview of a gene set observed to have a similar transcriptional response when upstream
genes are perturbed.
Input: You will be provided descriptions of the constituent genes. Ensure that your overview remains faithful to this
domain knowledge.
Output format: Fill in each of these three sections.
1) Brief overview of gene set: Write one to two sentences summarizing how the given genes are related. Focus on the
most specific pathways that are common among these genes.
2) Upstream pathways may affect this gene set: Write one to two sentences describing what types of perturbations might
impact the expression of the genes in this gene set. For example, you might consider pathways that are upstream of
these genes or compensatory mechanisms.
3) Name of gene set: Summarize the gene set within ten words.
Constraints:
- Maintain a professional tone throughout.
- Do not comment on your own writing.
- Do not add any notes or references. Do not make up additional information.
- Do not discuss the importance or impact of the gene set. Focus only on its function.
[End of Prompt]
[Start of Input] {list of gene summaries} [End of Input]
```

## C.2 QUESTION-ANSWERING TEMPLATES

For differential expression and direction of change, we used the following template for SUMMER.

[Start of Prompt]

You are an expert molecular biologist who studies how genes are related using Perturb-seq. Your goal is to determine: Does a CRISPRi knockdown of {perturbation} in {cell line} result in differential expression of {gene}?

You are given as input:

- Description of perturbed gene ({perturbation}): description of gene that is perturbed via CRISPRi knockdown

- Description of gene of interest ({gene}): description of gene, the impact on which you wish to infer

- Context: description of cell line in which the genes are expressed

- Examples: set of experimental observations that describe the impact of CRISPRi perturbations on related genes

Output: Please extract the most relevant parts of the examples that address these five questions. Be specific.

1) Which of the observed perturbed genes are most similar to {perturbation} (if any, including {perturbation} itself)?

2) When perturbing {perturbation} or similar genes, what downstream pathways or genes are differentially expressed? Justify your answer with the observed outcomes.

3) Which of the observed genes of interest are most similar to {gene} (if any, including {gene} itself)?

4) What perturbations of upstream pathways or genes result in differential expression of {gene} or similar genes (if any)? Justify your answer with the observed outcomes.

5) Is a CRISPRi knockdown of {perturbation} in {cell line} likely to result in differential expression of {gene}? For example, if 2) and 4) are unrelated or only indirectly related, it is unlikely we will observe differential expression. On the other hand, if 2) and 4) significantly overlap in specific genes or pathways, we may observe differential expression. Your final answer should end with one of these three options and nothing else.

- No. Knockdown of {perturbation} does not impact {gene}.

- Yes. Knockdown of {perturbation} results in differential expression of {gene}.

- There is insufficient evidence to determine how knockdown of {perturbation} affects {gene}.

[End of Prompt]

[Start of Input]

- Description of perturbed gene ({perturbation}): {summary of perturbation}

- Description of gene of interest ({gene}): {summary of downstream gene}

- Context: {sentence describing cell line}

- Examples: {list of examples}

[End of Input]

For direction of change, we change the answer options to the following.

A) Knockdown of {perturbation} results in a decrease in expression of {gene}.

B) Knockdown of {perturbation} results in an increase in expression of {gene}.

The following template was used for the LLM (No CoT) baseline on differential expression.

You are an expert molecular biologist who studies how genes are related using Perturb-seq.

You are given as Input:

- Perturbed gene: description of gene that is perturbed via CRISPRi knockdown

- Gene of interest: description of gene, the impact on which you wish to infer

Context: {sentence describing cell line}

Question: If you knockdown the perturbed gene using CRISPRi, how does the gene of interest's expression change?

Answer: Your answer must end with one of these two choices and nothing else.

A) Knockdown of the perturbed gene does not impact the gene of interest.

B) Knockdown of the perturbed gene results in differential expression of the gene of interest.

Format: Follow the same format as Examples 1 and 2, and complete Example 3.

Example 1.

Input:

- Perturbed gene: {summary of perturbation}

- Gene of interest: {summary of downstream gene}

Answer: {either A) ... or B) ...}

Example 2. {same format as Example 1, opposite Answer}

Example 3. {same format as Example 1, empty Answer}

The following template was used for the LLM (No retrieval) baseline on differential expression. Both answer options are provided twice each as hypotheses, regardless of the ground truth answer.

You are an expert molecular biologist who studies how genes are related using Perturb-seq.

You are given as Input:

- Perturbed gene: description of gene that is perturbed via CRISPRi knockdown

- Gene of interest: description of gene, the impact on which you wish to infer

- Hypothesis: hypothesis regarding how the specified perturbation affects the gene of interest

Context: {sentence describing cell line}

Question: If you knockdown the perturbed gene using CRISPRi, how does the gene of interest's expression change?

Task: Your goal is to identify evidence in the input that supports or refutes the hypothesis, and explain whether the hypothesis is likely to be true.

Output format: Please fill in the following four sections. Preserve the formatting and add the corresponding content.

1) Supporting evidence: Identify all relevant parts of the input that support the hypothesis.

2) Refuting evidence: Identify all relevant parts of the input that refute the hypothesis.

3) Explanation: Based on the evidence, explain how to answer the question, step by step. In particular,

- if there is a causal relationship from the perturbed gene to the gene of interest, explain how biological mechanisms relate the perturbed gene to the gene of interest.

- if there is no causal relationship from the perturbed gene to the gene of interest, explain why. For example, the perturbed gene may be downstream of the gene of interest, or there may be no relationship between the two genes.

- if there is insufficient evidence to answer the question, say so.

4) Answer: Your answer must end with one of these three choices and nothing else.

A) Knockdown of the perturbed gene does not impact the gene of interest.

B) Knockdown of the perturbed gene results in differential expression of the gene of interest.

C) There is insufficient evidence to determine how knockdown of the perturbed gene affects the gene of interest.

Input:

- Perturbed gene: {summary of perturbation}

- Gene of interest: {summary of downstream gene}

- Hypothesis: {either A) ... or B) ...}

For direction of change, we change the answer options to the following.

A) Knockdown of the perturbed gene results in down-regulation of the gene of interest.

B) Knockdown of the perturbed gene results in up-regulation of the gene of interest.

# D   EXAMPLES

Full LLM outputs are available in our data distribution. We sample real inputs/outputs for illustrative purposes.

## D.1   SUMMARIZATION EXAMPLES

Here is an example of the knowledge-base description for ABCE1, used as the {description} in the single-node summary.

Description of gene: ATP binding cassette subfamily E member 1

Gene products: ATP-binding cassette sub-family E member 1

Functions: (Microbial infection) May act as a chaperone for post-translational events during HIV-1 capsid assembly., (Microbial infection) Plays a role in the down-regulation of the 2-5A/RNASEL pathway during encephalomyocarditis virus (EMCV) and HIV-1 infections., Nucleoside-triphosphatase (NTPase) involved in ribosome recycling by mediating ribosome disassembly.

Quaternary structure: (Microbial infection) Interacts with HIV-1 proteins Vif and Gag. (Microbial infection) Interacts with HIV-2 protein Gag. Interacts with PINK1.

Interacts with: ABCE1, EIF3J, GAG_HV1B1, MATRX_CHAV, PHOSP_RABVE

Biological process: enables endoribonuclease inhibitor activity (Binds to and stops, prevents or reduces the activity of endoribonuclease.)

Here is the unedited output for the single-node summary, which is used as input to the 1-hop neighborhood summary.

The primary molecular and cellular function of gene ABCE1 is to facilitate ribosome recycling by mediating ribosome disassembly through its nucleoside-triphosphatase (NTPase) activity, and to act as a chaperone for post-translational events during HIV-1 capsid assembly. Additionally, ABCE1 plays a role in the down-regulation of the 2-5A/RNASEL pathway during viral infections. Perturbing gene ABCE1 via gene knockdown may disrupt ribosome recycling and viral capsid assembly, potentially leading to aberrant protein synthesis and altered viral replication dynamics.

Here is an example of the knowledge graph relationships for ABCE1.

Relations to other genes:

- ABCE1 enables 4 iron, 4 sulfur cluster binding, ATP binding, ATP hydrolysis activity, CTPase activity, GTPase activity, endoribonuclease inhibitor activity, iron ion binding, ribonucleoside triphosphate phosphatase activity, ribosomal small subunit binding.

- ABCE1 is active in cytosolic ribosome

- ABCE1 is involved in negative regulation of endoribonuclease activity, regulation of translation, rescue of stalled ribosome, ribosomal subunit export from nucleus, ribosome disassembly, translational initiation, translational termination, cytoplasm, cytosol, membrane, mitochondrial matrix, mitochondrion.

- Based on evidence from experimental evidence in humans, database evidence in humans, ABCE1 may physically interact with RNASEL.
- Based on evidence from experimental evidence in humans, experimental evidence in other animals, ABCE1 may physically interact with EIF1AX, EIF3A, EIF3B, EIF3C, EIF3D, EIF3E, EIF3F, EIF3G, EIF3H, EIF3I, EIF3K, EIF3L, EIF3M, G3BP2, LTO1, MFGE8, PSMD14, RACK1, RPL12, RPL23, RPL4, RPL7A, RPL9, RPL9P7, RPL9P8, RPL9P9, RPS10, RPS10-NUDT3, RPS11, RPS12, RPS13, RPS14, RPS15, RPS15A, RPS16, RPS17, RPS18, RPS19, RPS2, RPS20, RPS21, RPS24, RPS25, RPS26, RPS27, RPS27A, RPS28, RPS29, RPS3, RPS3A, RPS4X, RPS5, RPS6, RPS7, RPS8, RPS9, RPSA, YAE1.
- Based on evidence from experimental evidence in humans, experimental evidence in other animals, literature evidence in humans, ABCE1 may physically interact with EIF5, RPS23.
- Based on evidence from experimental evidence in humans, experimental evidence in other animals, literature evidence in humans, literature evidence in other animals, ABCE1 may physically interact with EIF1, ETF1.
- Based on evidence from experimental evidence in humans, experimental evidence in other animals, literature evidence in other animals, ABCE1 may physically interact with EIF3J, RPLP0.
- Based on evidence from experimental evidence in humans, literature evidence in humans, ABCE1 may physically interact with DCP2, EIF2S3, FDPS, PSMD1, PSMD4.
- Based on evidence from experimental evidence in humans, literature evidence in humans, literature evidence in other animals, ABCE1 may physically interact with EIF2S1, EIF2S2.
- Based on evidence from experimental evidence in other animals, literature evidence in humans, ABCE1 may physically interact with UBL4A.
- Based on evidence from experimental evidence in other animals, literature evidence in other animals, ABCE1 may physically interact with DENR, EIF1B, MRTO4.
- In 293T cells, ABCE1 may form a complex with IGF2R
- In the cytosol, ABCE1 enables OAS antiviral response, RNASEL binds 2'-5' oligoadenylate, ABCE1 binds RNASEL, OAS antiviral response, RNASEL binds 2'-5' oligoadenylate, ABCE1 binds RNASEL
- In the mitochondrial matrix, ABCE1 enables Interferon alpha/beta signaling, ABCE1 binds RNASEL, inhibiting it, Interferon alpha/beta signaling, ABCE1 binds RNASEL, inhibiting it

Here is the unedited output for the 1-hop neighborhood summary.

Perturbations that may impact the levels of ABCE1 include alterations in the 2-5A/RNASEL pathway, HIV-1 capsid assembly, or ribosome recycling, as well as changes in the interactions with its binding partners, such as HIV-1 proteins Vif and Gag, HIV-2 protein Gag, PINK1, EIF3J, and MATRX_CHAV. Additionally, perturbations in the regulation of translation, translational initiation, and translational termination may also affect ABCE1 levels. Furthermore, changes in the activity of its binding partners, such as EIF1AX, EIF3A, EIF3B, and RPL12, may also impact ABCE1 expression.

## D.2 QUESTION-ANSWERING EXAMPLES

Here is an example of the {list of examples} used in the QA template (C.2), retrieved to predict the impact of AARS2 knockdown on AAK1.

Example 1: Impact of knocking down MTIF2 on SPHK1

Description of perturbed gene (MTIF2): The primary molecular and cellular function of gene MTIF2 is to facilitate the initiation of protein synthesis in mitochondria by protecting formylmethionyl-tRNA from spontaneous hydrolysis and promoting its binding to the 30S ribosomal subunits, ultimately leading to the formation of the 70S ribosomal complex. This process is crucial for the translation of mitochondrial DNA into functional proteins. Perturbing gene MTIF2 via gene knockdown would likely disrupt the initiation of protein synthesis in mitochondria, potentially leading to impaired mitochondrial function and protein synthesis.

Description of gene of interest (SPHK1): The primary molecular and cellular function of gene SPHK1 is to catalyze the phosphorylation of sphingosine to form sphingosine 1-phosphate, a lipid mediator with both intra- and extracellular functions, and to interact with various proteins to regulate downstream cellular processes. Additionally, SPHK1 plays a role in the sphingosine-1-phosphate receptor signaling pathway, which regulates transcription and other cellular processes. Perturbations that might impact the expression of gene SPHK1 include alterations in the sphingosine-1-phosphate receptor signaling pathway, changes in calmodulin binding, or disruptions to its interactions with other proteins such as ACY1, CATB, EF1A1, FHL2, LAMP2, SHLB1, SPHKAP, or TRAF6.

Outcome: A) Knockdown of MTIF2 does not impact SPHK1.

Example 2: Impact of knocking down CLPX on PTCD1

Description of perturbed gene (CLPX): The primary molecular and cellular function of gene CLPX is to act as an ATP-dependent specificity component of the Clp protease complex, hydrolyzing ATP and forming a homohexamer ring structure that assembles with CLPP rings to form the Clp complex. This complex is involved in protein degradation and quality control in the mitochondrial matrix. Perturbing gene CLPX via gene knockdown may disrupt the proper functioning of the Clp protease complex, leading to impaired protein degradation and potential accumulation of misfolded or damaged proteins in the mitochondrial matrix.

Description of gene of interest (PTCD1): The primary molecular and cellular function of gene PTCD1 is to negatively regulate leucine tRNA levels, mitochondria-encoded proteins, and COX activity, while also affecting the 3'-processing of mitochondrial tRNAs, thereby influencing mitochondrial protein synthesis. As a mitochondrial protein, PTCD1 associates with mitochondrial leucine tRNAs and interacts with various proteins, including ELAC2, to modulate its functions. Perturbations that might impact the expression of gene PTCD1 include disruptions to mitochondrial tRNA metabolism, alterations in COX activity, or changes in the levels of interacting proteins, such as ELAC2, which could in turn affect PTCD1's regulatory roles in mitochondrial protein synthesis.

Outcome: A) Knockdown of CLPX does not impact PTCD1.

{...}

Example 12: Impact of knocking down NARS2 on REPS1

Description of perturbed gene (NARS2): The primary molecular and cellular function of gene NARS2 is to encode a mitochondrial aminoacyl-tRNA synthetase that catalyzes the attachment of asparagine to its cognate tRNA, thereby participating in protein synthesis. This enzyme enables asparagine-tRNA ligase activity, facilitating the formation of Asn-tRNA(Asn) from L-asparagine and tRNA(Asn). Perturbing gene NARS2 via gene knockdown may disrupt the proper charging of tRNA with asparagine, potentially leading to aberrant protein synthesis and mitochondrial dysfunction.

Description of gene of interest (REPS1): The primary molecular and cellular function of gene REPS1 is to coordinate the cellular actions of activated EGF receptors and Ral-GTPases, potentially linking them to an EGF-responsive tyrosine kinase through its interactions with CRK and GRB2. Additionally, REPS1 may be involved in clathrin-mediated endocytosis through its interactions with AMPH, ITSN1, and SGIP1. Perturbations to the EGF signaling pathway, Ral-GTPase activity, or clathrin-mediated endocytosis may impact the expression of gene REPS1.

Outcome: A) Knockdown of NARS2 does not impact REPS1.

Here is the unedited output for the above prompt.

[Start of Output]

Based on the provided examples, I will extract the most relevant parts to address the five questions.

1. Which of the observed perturbed genes are most similar to AARS2 (if any, including AARS2 itself)?

From the examples, we can see that AARS2 is a mitochondrial aminoacyl-tRNA synthetase, specifically responsible for attaching alanine to tRNA(Ala). The most similar genes to AARS2 are likely to be other mitochondrial aminoacyl-tRNA synthetases, such as TARS2 (threonine-tRNA ligase), RARS2 (arginine-tRNA ligase), and NARS2 (asparagine-tRNA ligase). These genes share similar functions and are also involved in protein synthesis.

2. When perturbing AARS2 or similar genes, what downstream pathways or genes are differentially expressed? Justify your answer with the observed outcomes.

From the examples, we can see that perturbing AARS2 or similar genes does not impact the expression of downstream genes. For instance, knocking down AARS2 does not affect the expression of SPHK1, PTCD1, SLC35G2, LYRM7, RTN2, AAK1, or NECAP2. Similarly, knocking down TARS2, RARS2, or NARS2 does not impact the expression of SLC35G2, REPS1, or NECAP2, respectively. This suggests that the perturbation of AARS2 or similar genes does not significantly affect downstream pathways or genes.

3. Which of the observed genes of interest are most similar to AAK1 (if any, including AAK1 itself)?

From the examples, we can see that AAK1 is a gene involved in regulating clathrin-mediated endocytosis and interacts with clathrin and other proteins. The most similar genes to AAK1 are likely to be other genes involved in clathrin-mediated endocytosis, such as NECAP2, which facilitates endocytosis through its interactions with adapter protein complexes AP-1 and AP-2. Other genes that interact with clathrin, such as RTN2, may also be similar to AAK1.

4. What perturbations of upstream pathways or genes result in differential expression of AAK1 or similar genes (if any)? Justify your answer with the observed outcomes.

From the examples, we can see that perturbing genes involved in clathrin-mediated endocytosis, such as NECAP2, does not impact the expression of AAK1. Similarly, perturbing genes involved in protein synthesis, such as RPL10, does not affect the expression of AAK1. This suggests that the perturbation of upstream pathways or genes does not significantly affect the expression of AAK1 or similar genes.

5. Is a CRISPRi knockdown of AARS2 in K562 cells likely to result in differential expression of AAK1?

Based on the observed outcomes, it is unlikely that a CRISPRi knockdown of AARS2 in K562 cells will result in differential expression of AAK1. The perturbation of AARS2 or similar genes does not impact the expression of downstream genes, including AAK1. Therefore, the answer is:

No. Knockdown of AARS2 does not impact AAK1.

[End of Output]

## D.3 OTHER EXAMPLES

We use the following for {sentence describing cell line}, derived from Wikipedia.

K562 cells are immortalised myelogenous leukemia cells of the erythroleukemia type.

RPE1 cells are a non-cancerous, hTERT-immortalized, near-euploid, adherent, and p53-positive cell line.

Jurkat cells are an immortalized line of human T lymphocyte cells.

HepG2 cells are a human liver cancer cell line, derived from a patient with a well-differentiated hepatocellular carcinoma.

