# OpenReview forum: "Contextualizing biological perturbation experiments through language"
_ICLR.cc/2025/Conference — ICLR 2025 Poster_

### Official Review · Reviewer_qAqz · 2024-10-29

**Soundness:** 2
**Presentation:** 1
**Contribution:** 2
**Rating:** 3
**Confidence:** 5

**Summary:**

This paper proposed an in-silico approach to solving the biological perturbation experiment results prediction. The original idea is to enrich the graph-structure-based cellular responses textually. In terms of technique details, the authors proposed PertubQA as a pre-processing step and omitted minor changes in the gene expression level. Then, the authors adopted prompt engineering to retrieve the structured knowledge crossing gene-gene interaction result and gene description. Following a CoT manner, the LLM generates the final answers.

**Strengths:**

S1. Adopting LLM in biological analysis is an interesting topic. This study follows a combination of LLM-related tech fashion.

S2. The authors developed a PertubQA dataset to validate the following research on textual enrichment of perturbation analysis.

S3. Detailed examples and case studies are provided to show the effectiveness of the framework.

**Weaknesses:**

W1. Part of the technical contribution is related to GenePT, which adopted a textual description and gene expression value to extract the cell representation, and GEARs, which combined prediction on a genetic relation graph (gene-coexpression graph and GO graph).

W2. The contributions of the proposed PerturbQA dataset are unclear. Why do we need a vague textual description of gene perturbation instead of a formalized graph structure with exactly numeric data?

W3. Missing technical details. I couldn't find the setting (or formal definition) of the study problem. Those make this paper hard to follow.

**Questions:**

Q1. Please describe the distinct contributions of PerturbQA beside a RAG- or a meta-path-walk-like text description generation. How do you ensure the description is correct? For instance, PubmedQA is generated from many articles and certified by human experts.

For the rest of the question, please refer to weak points.

---

> ### Author Response · Authors · 2024-11-20
> **Thank you for your review**
>
> Thank you for your review and suggestions! We hope this response provides clarity regarding your concerns, and we look forward to discussing.
>
> ## Contribution and data quality
>
> PerturbQA is first and foremost, a carefully curated *benchmark* for language-based reasoning for single-cell perturbations.
>
> Compared to existing benchmarks like PubmedQA, the ground truth is not determined by fact checking, but by experimental assays. The quality of our labels can be assessed based on statistical consistencies. Figure 4 illustrates that the Wilcoxon ranked-sum test is reasonably calibrated on our datasets (for determining differential expression), and Figures 5+6 illustrate that in two near-biological replicates, there is high agreement. We also employ conservative thresholds for selecting positives and negatives (A.2), and exclude statistically uncertain examples.
>
> The textual descriptions extracted from knowledge graphs are manually curated by large consortia, compiling decades of research. We believe that the identifier mapping tables they maintain offer a much cleaner means of extracting information relevant to each gene, compared to search engine / embedding-based approaches more common in NLP. While we cannot guarantee that the knowledge is correct, we do include context regarding data provenance (cell line, assay type, whatever is available), as LLMs have demonstrated the ability to filter noisy information, when this is provided [1].
>
> Finally, we do not claim that the LLM-generated summaries are correct; only that they are helpful for making the end prediction, on which the framework is evaluated.
>
> [1] Allen-Zhu and Li. Physics of Language Models: Part 3.3, Knowledge Capacity Scaling Laws. 2024.
>
> ## Algorithmic novelty
>
> Please see the general response. To summarize: applying NLP techniques out of the box results in near-random performance on PerturbQA, and they must be adapted in domain-specific ways. To demonstrate that this task is feasible, we introduced a *minimal* LLM-based example (SUMMER), which draws from common LLM techniques. We do not claim that this method is novel from an NLP perspective. We *do* claim that this LLM-reasoning based framework is novel in context of perturbation modeling.
>
> ## Specific questions
>
> **why not "formalized graph structure with exactly numeric data":**
>
> Both GEARS and GAT operate over formal graph structures. GEARS leverages real-valued expression matrices, while GAT is trained on our discretized labels. Both methods underperform in Table 1, and this can be attributed to two factors (Section 3, "modeling perturbations").
>
> - "Raw" single cell expression matrices are actually the end product of extensive preprocessing pipelines, and they are subject to high aleatoric and epistemic noise (batch effects) [2]. In fact, even two datasets generated by the same lab, of the same cell line, can be largely inconsistent at the individual gene level [3]. This is why we employed stringent criteria on PerturbQA examples (A.2), and why we choose to interpret single-cell perturbation data at the level of discrete insights, in line with best practices (Section 3, "statistical insights").
>
> - Biological knowledge graphs are highly heterogeneous, spanning [many layers of hierarchy](https://geneontology.org/docs/ontology-documentation/), and are annotated with details regarding the context and provenance of each observation. When current graph-based methods translate annotations into adjacencies, the semantics of each relationship are also lost: edges like "enables," "does not enable," "is part of" are all mapped to "1." Finally, different knowledge graphs operate at differing levels of quality/noise, making their harmonization difficult. Therefore, we only use the graph structure to inform the retrieval and summarization of relevant information, rather than as a strict backbone to constrain modeling.
>
> [2] Luecken and Theis. Current best practices in single‐cell RNA‐seq analysis: a tutorial. Mol Syst Biol (2019) 15: e8746.
>
> [3] Nadig et al. Transcriptome-wide characterization of genetic perturbations. 2024.
>
> **formal task definitions:** We have included formal definitions for each task in Section 4.1, with motivation in Section 3. Could you please let us know which specific aspects of the notation / setting you find confusing, so we can try to improve the presentation? Thank you!
>
> **GEARS and GenePT:** We evaluate GenePT and GEARS on our benchmark for completeness, as they represent state-of-the-art approaches for this task. We do not consider their adaptations (for discrete classification) part of our primary technical contribution.

---

> > ### Comment · Reviewer_qAqz · 2024-11-26
> >
> > After checking the rebuttal and the rebuttal to all reviewers, I remain concerned about the application scenario of the proposed dataset, PerturbQA. I insist to keep my evaluation.

---

> > > ### Author Response · Authors · 2024-11-26
> > > **Thank you for your response**
> > >
> > > Thank you for your response and for reading our rebuttals! Could you clarify what aspects of the application remain concerning, and which aspects of the task definitions are unclear? This would be very helpful for improving our paper.

---

### Official Review · Reviewer_ii1Q · 2024-10-31

**Soundness:** 3
**Presentation:** 3
**Contribution:** 2
**Rating:** 6
**Confidence:** 4

**Summary:**

The paper introduces PERTURBQA, a benchmark for using language models to interpret genetic perturbation experiments. These experiments reveal gene functions but are expensive. The proposed SUMMER framework combines biological text data and experimental results, outperforming knowledge-graph-only models on tasks like predicting gene expression changes and gene set enrichment. SUMMER’s language-based outputs offer clearer insights for biologists, enhancing interpretability and accessibility in modeling biological data.

**Strengths:**

The paper is well-organized and easy to follow.

The paper proposed an interesting problem of using LLMs to predict gene pertubation.

**Weaknesses:**

1. The biggest concern of the proposed method is the problem itself. For example, SUMMER relies on the biology knowledge graph. However, the biology knowledge graph is built based on the experiments and analysis. Compared with the gene perturbation prediction from scRNA, the knowledge-building quiring updates the knowledge at once to build the possible links among genes. This indicates that this method has a risk of overfitting the known knowledge and being unable to discover new perturbation genes; it is more like a retrieval system for gene interactions we already know.

To further explain how the proposed method can generate more situations, my suggestion is to include more datasets to avoid overfitting. For example, in scGPT[1], the evaluation for perturbation is across 3 different perturbation datasets.

2. In the previous study of gene perturbation, one gene input can output several gene expressions, which are judged at one time. However, the LLM framework suggests that the source and target are a pair of data. This means that if we want to see every gene expression after perturbation, we should go through every combination of gene pairs with LLMs. This is really time-consuming in this setting. Besides, the experiments are really limited to a small size of genes.

In GEARS, scGPT, and CellPLM, they set the perturbation tasks for both 1 gene unseen and 2 genes unseen, and they can claim that the proposed method can handle different perturbation situations. However, in this paper, there are no such illustrations. Besides, in GEARS, the authors predict 102 genes at one time for novel gene perturbation analysis (Fig5a in GEARS). In scGPT, they predict 210 combinations (Fig3 in scGPT). In this paper, there is no evidence that the proposed method can do such things.

3. The authors should include gene pretrained methods such as scGPT[1] and Geneformer[2]. The text-based methods are pretrained on lots of textual information, the comparing method is unfair with only MLP, GAT, and GEARS.

As I mentioned before, the pretrained methods on the single-cell data are considered one of the most powerful methods in gene perturbation tasks at this moment. In scGPT, they report scGPT can achieve 50% higher than GEARS. Instead of training on lots of textual information, they trained directly from gene expressions. If the authors want to claim the textual-based pretrained method is more powerful than we predict genes from scRNA data, they need to discuss these models.

4. Compared with other work in gene expression from ML method, such as CellPLM[3], LanCell[4], CellSentence[5] this paper lacks biology analysis in bio view. This limited the application to further applications.

In scGPT and GEARS, they have a figure of predicted gene expression profiles of the perturbation conditions. Other papers are not perturbation-specific methods, but they are ai4biology papers that are published in the top AI conferences. They also show some biological analysis to strengthen their methods for their own contribution, such as marking genes on the cells to show the cell states.

[1] Cui H, Wang C, Maan H, et al. scGPT: toward building a foundation model for single-cell multi-omics using generative AI[J]. Nature Methods, 2024: 1-11.

[2] Theodoris C V, Xiao L, Chopra A, et al. Transfer learning enables predictions in network biology[J]. Nature, 2023, 618(7965): 616-624.

[3] Wen H, Tang W, Dai X, et al. CellPLM: Pre-training of Cell Language Model Beyond Single Cells[C]//The Twelfth International Conference on Learning Representations.

[4] Zhao S, Zhang J, Wu Y, et al. LangCell: Language-Cell Pre-training for Cell Identity Understanding[C]//Forty-first International Conference on Machine Learning.

[5] Levine D, Rizvi S A, Lévy S, et al. Cell2Sentence: Teaching Large Language Models the Language of Biology[C]//Forty-first International Conference on Machine Learning.

**Questions:**

see weakness

---

> ### Author Response · Authors · 2024-11-20
> **Thank you for your review (1/2)**
>
> Thank you for your review and questions. We hope this response clarifies certain points of confusion, and we look forward to discussing with you!
>
> ## Retrieval vs. discovery
>
> **"retrieval system:"** Please see the general response. To summarize: Only ~3% of our testing pairs physically interact in any context, while only ~20% of pairs are directly connected by any annotation (including coarse ones). We find that the presence of physical interactions is minimally predictive of differential expression (DE).
>
> **"my suggestion is to include more datasets [...] For example, in scGPT, the evaluation for perturbation is across 3 different perturbation datasets"**
>
> In scGPT, the evaluation was conducted over 3 experiments in the **same K562 cell line**, including the K562 essential screen that we also incorporate. The other two are older (2016, 2019) and much smaller, as single-cell CRISPR technologies have become more scalable and efficient in the past 5 years.
>
> In contrast, our experiments cover **4 cell lines, derived from 5 experiments**, which are the **largest publicly-available Perturb-seq screens** [1,2]. Biologically, cell lines are very distinct, and they are derived from different cancer tumors / other conditions (K562 myelogenous leukemia, RPE non-cancerous, HepG2 liver cancer, Jurkat acute T cell leukemia). In particular, different genes are expressed, and among the same genes, the marginal distributions (of expression) and gene-gene relationships may differ. To ensure the quality and consistency of our benchmark, we have chosen to focus on these 5 screens, which are larger and published more recently.
>
> [1]. Replogle et al. Mapping information-rich genotype-phenotype landscapes with genome-scale Perturb-seq. Cell. 2022.
>
> [2] Nadig et al. Transcriptome-wide characterization of genetic perturbations. 2024.
>
> ## Combinatorial perturbations
>
> The lack of combinatorial perturbations is less a limitation of the method, but of evaluation.
>
> To craft a "proof of concept" for combinatorial perturbations, it would be relatively straightforward to summarize the neighborhoods of each participating gene, and prompt for any synergy/lack thereof.
>
> However, existing datasets for combinatorial perturbations are limited. The most commonly analyzed are Norman et al. [3] and Wessels et al. [4], each containing around 100 perturbation pairs. These two experiments operate in different modalities (CRISPR activation vs. inhibition), and since there are no alternatives for comparison, it is difficult to quantify the quality of these data. A major goal of this work was to craft a trustworthy benchmark for perturbation modeling, so we chose to focus on single perturbations instead. Therefore we consider combinations currently out of scope, but an opportunity for future work when better datasets are available.
>
> [3] Norman et al. Exploring genetic interaction manifolds constructed from rich single-cell phenotypes. Science. 2019.
>
> [4] Wessels et al. Efficient combinatorial targeting of RNA transcripts in single cells with Cas13 RNA Perturb-seq. Nat Methods. 2023.

---

> > ### Author Response · Authors · 2024-11-20
> > **Thank you for your review (2/2)**
> >
> > ## Additional biological analysis
> >
> > Thank you for the recommendation! We have updated the draft with several additional analyses regarding the qualitative aspects of our framework (currently in Appendix C), including evaluation from a domain expert. Here is a summary.
> >
> > - Clusters that elude manual annotation tend to be smaller or exhibit lower agreement. On these, gene set over-representation analysis focuses on highly specific gene sets, which each cover subsets of these clusters, rather than the whole. The LLM takes the opposite approach, and its summaries tend to "lift" the description to higher levels of hierarchy (Table 8). The two strategies provide orthogonal information, though the LLM outputs may be more readable.
> >
> > - We analyzed 300 generations (3 trials of 100 DE examples) to understand common failure modes (detailed examples in C.3). Errors and inconsistencies primarily resulted from deductions backed by overly-generic information. For example, the LLM may list an excessively broad set of influences, e.g. "mitochondrial function, protein synthesis, or transcriptional regulation," which affect nearly everything.
> >
> >   In several instances, the LLM was also confused between concepts which may be loosely connected, but not in the same context. For example, Gene A is upstream of stress signaling, e.g. "oxidative stress," which is related to the mitochondria. However, Gene A is *not* responsible for healthy mitochondria function, and should not respond similarly to genes that are.
> >
> > - Finally, since this paper focuses on providing value to biologists, we recruited a domain specialist (molecular biologist, trained in wet lab and computational biology) for this task (See C.1).
> >
> > Overall, the LLM-generated summary was equal or more informative than the classical gene set enrichment results in 92% of cases, and agrees with the independent annotator in 72% of cases.
> >
> > 1. In 21/25 cases, the biologist reported that the LLM-generated summary was more informative. In 2/25 cases, they contained the same amount of information; and in 2/25 cases, the gene set contained more information.
> > 2. In 18/25 cases, the biologist reported that the LLM summary captured the same biology as the original human annotation (our ground truth labels).
> >
> > 1. In the 2 cases where the gene set contained more *information*, a list of specific protein complexes were discovered, e.g.
> >     ```
> >     EIF2AK4 (GCN2) binds tRNA, Aminoacyl-tRNA  binds to the ribosome at the A-site, 80S:Met-tRNAi:mRNA:SECISBP2:Sec-tRNA(Sec):EEFSEC:GTP is hydrolysed to 80S:Met-tRNAi:mRNA:SECISBP2:Sec and EEFSEC:GDP by EEFSEC, UPF1 binds an mRNP with a termination codon preceding an Exon Junction Complex, Translocation of ribosome by 3 bases in the 3' direction, Translation of ROBO3.2 mRNA initiates NMD.
> >     ```
> >     However, this information-rich output is hard to interpret, compared to the LLM output, which the annotator marked as agreeing with the label of "translation."
> >     ```
> >    Ribosomal Protein Components Involved in Translation: This gene set is comprised of components of the large and small ribosomal subunits, which are essential for protein synthesis and translation. These genes are involved in the assembly and function of the ribosome, facilitating the translation of messenger RNA into protein.
> >    ```
> >
> > 2. In the 7/25 cases where the LLM summary differed from the human annotation, the LLM annotation tended to miss some high specific terms, e.g. "targets of nonsense-mediated decay" was generalized to "stress response," and "dysregulated lncRNA antisense transcripts" was generalized to "nuclear gene regulation." Related terms tend to be sparsely annotated in Gene Ontology, so this indicates that it would be useful to tune the granularity of generations in the future, or to generate multiple candidates for specific descriptions.
> >
> > **scGPT:** We finetuned scGPT + GEARS using their published "perturbation" tutorial. Due to time constraints, we were only able to benchmark K562, but we will add the full results to the final paper. We will also include discussion of these references in our work. Thank you!
> >
> > scGPT + GEARS averaged 0.52 AUC on K562 (compared to SUMMER 0.6, GenePT 0.57). Similar to other methods, this model predicts "perfectly" on some perturbations (104/267 with AUC=1) while guessing randomly on the remaining, so it appears that either the embeddings are unhelpful, or the GEARS backbone (on which scGPT builds) is suboptimal.

---

> ### Comment · Reviewer_ii1Q · 2024-11-24
>
> Thank you for your response. Personally, my main concern is how this method influences perturbation analysis from both a logical and practical perspective. I believe the perturbation problem is a complex process in the biological domain. As you mentioned, even the identified perturbation pairs may not accurately represent the actual perturbations. With this in mind, I am unsure whether the knowledge derived is truly reliable for the perturbation task.
>
> One advantage of using pretrained models in perturbation analysis is their ability to study RNA data without needing to verify the knowledge's accuracy.  In practice, I also think the pretrained single-cell models need less data to finetune, or infer, or predict the perturbation combination. Additionally, it’s worth noting that scGPT only extracts matching genes from GEARS for specific datasets rather than utilizing GEARS’ structure. Therefore, the results based on the GEARS backbone may not fully align with the implementation.
>
> I appreciate this novel perspective on the perturbation task, particularly how it combines single-cell problems with LLMs, and I will adjust my score because of this novelty.

---

> > ### Author Response · Authors · 2024-11-24
> > **Thank you for your response!**
> >
> > Thank you for your response and for updating your score!
> >
> > We agree that scGPT and other single cell foundation models could capture orthogonal information, and we will update our paper with this discussion / error analysis (when full results are available).
> >
> > We did run the published scGPT codebase for perturbations. Upon closer inspection, we realize that the "graph" part of each "graph" is discarded and only the expression is retained, so the results should be faithful.

---

### Official Review · Reviewer_psYt · 2024-11-03

**Soundness:** 3
**Presentation:** 3
**Contribution:** 2
**Rating:** 5
**Confidence:** 3

**Summary:**

This paper introduces PERTURBQA, a benchmark suite for evaluating LLMs in reasoning over structured biological data from genetic perturbation experiments. The paper also presents SUMMER, a language-based framework designed to predict differential expression and direction of gene changes as well as to perform gene set enrichment. SUMMER combines knowledge graphs and retrieval-augmented generation to enhance interpretability, matching or surpassing state-of-the-art models on PERTURBQA tasks. The benchmark is designed to help researchers interpret outcomes in high-content genetic perturbation experiments and understand model limitations.

**Strengths:**

1. **Novel Application of Language-Based Reasoning in Pertburbation Task:** The paper offers a novel application of language-based reasoning to biological data, allowing PERTURBQA tasks to be approached in an interpretable way that benefits domain experts.
2. **Comprehensive Benchmark Design:** PERTURBQA includes real-world tasks relevant to differential expression, gene direction, and gene set enrichment, providing a holistic assessment of model reasoning on biological data.
3. **Interpretable Model Outputs:** SUMMER’s use of knowledge graphs and retrieval-augmented generation produces outputs that domain experts can readily interpret, addressing the limitations of black-box models in biological contexts.

**Weaknesses:**

1.**Insufficient Related Work Discussion:** The paper could better situate its contribution within the existing literature, especially regarding related work on several aspects: 1. graph-to-text works: such as [Zhao et al., 2023](https://arxiv.org/abs/2310.01089), [Chen et al., 2023](https://arxiv.org/abs/2307.03393), check the [survey](https://arxiv.org/pdf/2407.06564) for details. 2. graph RAG works, e.g. [GraphRAG](https://arxiv.org/pdf/2404.16130), [He et al., 2024](https://arxiv.org/abs/2402.07630), [Mavromatis et al., 2024](https://arxiv.org/abs/2405.20139) . Discussing these works would strengthen the contextual grounding of the proposed approach.

2.**Marginal Technical Contribution:** As stated in W1, both ideas of using LLM for graph tasks and graph RAG are widely studied before, which renders the contribution of this paper marginal.

3.**Limited Baselines:** A wider range of recent baselines, especially from the graph domain (e.g. those mentioned above in W1 and W2), should be included to provide a more comprehensive evaluation and enable better comparison of SUMMER's effectiveness.

4.**Potential Data Leakage Concerns:** Given that LLaMA3 might have been exposed to substantial amounts of biological data, including related gene interactions, there’s a risk of data leakage. A clearer evaluation of SUMMER’s performance independent of potential pre-trained biases could help clarify if its good performance stems from model design or from pre-existing knowledge in the model.

**Questions:**

NA

---

> ### Author Response · Authors · 2024-11-20
> **Thank you for your review**
>
> Thank you for your review and suggestions! We hope this response clarifies some of your confusion, and we look forward to discussing.
>
> **Literature review**: We are happy to expand our literature review and include a wider range of work on the NLP side. Thank you for the recommendations!
>
> **Technical contribution**: Please see the general response. To summarize: Our primary contribution is PerturbQA, a carefully crafted benchmark, derived from experimental assays and open knowledge graphs. Applying NLP techniques out of the box results in near-random performance on PerturbQA, and they must be adapted in domain-specific ways. To demonstrate that this task is feasible, we introduced a *minimal* LLM-based example (SUMMER), which draws from common LLM techniques. We do not claim that this method is novel from an NLP perspective. We *do* claim that this LLM-reasoning based framework is novel in context of perturbation modeling.
>
> **Additional baselines**: Please see the general response. To summarize: As Table 1 demonstrates, directly applying NLP methods out-of-the-box on PerturbQA is comparable to random guessing. It is *non-trivial* to adapt these methods for molecular biology reasoning, especially because the vast majority of biological literature is behind paywalls (vs. in machine learning, where everything is open).
>
> **Data leakage**: Please see the general response. To summarize: Our "no CoT" and "no retrieve" ablations performed near-random, so it appears that the base Llama model has very poor understanding of experimental outcomes. Furthermore, the gene pairs are also minimally represented in the knowledge graphs, and the presence of a known physical interaction is not predictive.

---

> ### Comment · Reviewer_psYt · 2024-11-25
> **Response to Rebuttal**
>
> Thank you for the effort put into the rebuttal.
>
> However, my primary concerns regarding the discussion of graph literature remain unresolved, as these works were neither compared nor even discussed in the updated manuscript.
>
> As a result, I will maintain my negative score.

---

> > ### Author Response · Authors · 2024-11-25
> > **Thank you for your response**
> >
> > Thank you for your response! We have updated the manuscript with your recommendations and additional relevant work (highlighted in blue on page 3).
> >
> > Currently we do not have the compute to run additional LLM-based experiments before the paper edit deadline, but please let us know if this addresses your concerns in part.

---

> > > ### Comment · Reviewer_psYt · 2024-11-26
> > > **Response to Rebuttal**
> > >
> > > Thanks for the prompt response and revision of the paper. However, I checked the updated related work and still believe that these methods weaken the technical contribution of the proposed method. Hence, I will maintain my original score.

---

> > > > ### Author Response · Authors · 2024-11-27
> > > > **Thank you for your response!**
> > > >
> > > > Thank you for your feedback! Please let us know if there's anything else we can do to improve the paper.
> > > >
> > > > Finally, we would like to reiterate that the primary novelty of this work is not the proposed algorithm, but the framework for modeling perturbation experiments. As we write in the Introduction (Paragraph 2) and Background (Section 3), current approaches for modeling perturbations are misaligned with what biologists aim to glean from these large-scale screens. We are the first to propose that predicting differentiation expression / direction of change and summarizing gene sets are more realistic endpoints, compared to existing regression-based objectives.
> > > >
> > > > Our hypothesis was simply, that language can be helpful for modeling perturbations in this setting. There are many approaches in modern NLP that could be adapted towards perturbation modeling, and we have prepared and presented perturbation screens / relevant knowledge sources in approachable formats for this purpose. We hope that this work can this work can encourage and facilitate such exploration.

---

### Official Review · Reviewer_2VEF · 2024-11-03

**Soundness:** 2
**Presentation:** 3
**Contribution:** 2
**Rating:** 3
**Confidence:** 3

**Summary:**

This paper introduces PERTURBQA, focusing on prompting LLMs for gene perturbation and gene set enrichment. It also presents a reasoning method, SUMMER, designed for this task. Experiments demonstrate the effectiveness of the proposed approach.

**Strengths:**

The proposed task of using large language models (LLMs) to address gene-related tasks is innovative and engaging.

The paper is clearly presented, well-written, and easy to follow.

**Weaknesses:**

Limited Experimental Insight: The experiments primarily conclude that SUMMER outperforms baselines but provide few additional insights into the new task. Given the novelty of using LLMs for this type of gene analysis, more extensive experiments and in-depth analysis would be valuable.

Insufficient Baselines and Model Comparisons: The baselines selected are not sufficiently comprehensive. Numerous studies have explored LLM reasoning with graph data or text retrieval, which are closely related to this method. Including these in the comparisons could yield deeper insights into the effectiveness of LLMs for gene-related tasks.

Metrics for Gene Set Enrichment: The suitability of ROUGE-1 recall and BERT Score for measuring the accuracy of gene set enrichment results is questionable. Human evaluation or evaluations with LLMs may offer more reliable assessments.

**Questions:**

Retrieved Content for Reasoning: Is the retrieved content primarily focused on the gene's function? If so, could this lead to information leakage in the question? For example, if asked about the influence of gene A on gene C, and the retrieved content on gene A directly states that A turn on C, wouldn’t this turn the task into reading comprehension rather than genuine prediction?

Multi-Hop Reasoning: How does SUMMER handle multi-hop reasoning if it only retrieves information on the one hop neighbors of the perturbation and target gene?

---

> ### Author Response · Authors · 2024-11-20
> **Thank you for your review (1/2)**
>
> Thank you for your review and recommendations! We hope this response provides clarity regarding your questions, and we look forward to discussing.
>
> ## Experimental insight
>
> Thank you for the recommendation! We have updated the draft with several additional analyses regarding the qualitative aspects of our framework (currently in Appendix C), including evaluation from a domain expert (further below). Here is a summary.
>
> - Clusters that elude manual annotation tend to be smaller or exhibit lower agreement. On these, gene set over-representation analysis focuses on highly specific gene sets, which each cover subsets of these clusters, rather than the whole. The LLM takes the opposite approach, and its summaries tend to "lift" the description to higher levels of hierarchy (Table 8). The two strategies provide orthogonal information, though the LLM outputs may be more readable.
>
> - We analyzed 300 generations (3 trials of 100 DE examples) to understand common failure modes (detailed examples in C.3). Errors and inconsistencies primarily resulted from deductions backed by overly-generic information. For example, the LLM may list an excessively broad set of influences, e.g. "mitochondrial function, protein synthesis, or transcriptional regulation," which affect nearly everything.
>
>   In several instances, the LLM was also confused between concepts which may be loosely connected, but not in the same context. For example, Gene A is upstream of stress signaling, e.g. "oxidative stress," which is related to the mitochondria. However, Gene A is *not* responsible for healthy mitochondria function, and should not respond similarly to genes that are.
>
> ## Human evaluation
>
> We are happy to provide human evaluation and will update the manuscript accordingly (See C.1). We recruited a domain specialist (molecular biologist, trained in wet lab and computational biology) for this task.
>
> Overall, the LLM-generated summary was equal or better to the classical gene set enrichment results in 92% of cases, and agrees with the independent annotator in 72% of cases.
>
> 1. In 21/25 cases, the biologist reported that the LLM-generated summary was more informative. In 2/25 cases, they contained the same amount of information; and in 2/25 cases, the gene set contained more information.
> 2. In 18/25 cases, the biologist reported that the LLM summary captured the same biology as the original human annotation (our ground truth labels).
>
> **Error analysis of human annotation:**
> 1. In the 2 cases where the gene set contained more *information*, a list of specific protein complexes were discovered, e.g.
>     ```
>     EIF2AK4 (GCN2) binds tRNA, Aminoacyl-tRNA  binds to the ribosome at the A-site, 80S:Met-tRNAi:mRNA:SECISBP2:Sec-tRNA(Sec):EEFSEC:GTP is hydrolysed to 80S:Met-tRNAi:mRNA:SECISBP2:Sec and EEFSEC:GDP by EEFSEC, UPF1 binds an mRNP with a termination codon preceding an Exon Junction Complex, Translocation of ribosome by 3 bases in the 3' direction, Translation of ROBO3.2 mRNA initiates NMD.
>     ```
>     However, this information-rich output is hard to interpret, compared to the LLM output, which the annotator marked as agreeing with the label of "translation."
>
>     ```
>    Ribosomal Protein Components Involved in Translation: This gene set is comprised of components of the large and small ribosomal subunits, which are essential for protein synthesis and translation. These genes are involved in the assembly and function of the ribosome, facilitating the translation of messenger RNA into protein.
>    ```
>
> 2. In the 7/25 cases where the LLM summary differed from the human annotation, the LLM annotation tended to miss some high specific terms, e.g. "targets of nonsense-mediated decay" was generalized to "stress response," and "dysregulated lncRNA antisense transcripts" was generalized to "nuclear gene regulation." Related terms tend to be sparsely annotated in Gene Ontology, so this indicates that it would be useful to tune the granularity of generations in the future, or to generate multiple candidates for specific descriptions.

---

> > ### Author Response · Authors · 2024-11-20
> > **Thank you for your review (2/2)**
> >
> > ## Other questions
> >
> > **Multi-hop reasoning:**
> > - Due to the hierarchical nature of the knowledge graphs under consideration (e.g. [the GO hierarchy](https://geneontology.org/docs/ontology-documentation/)), "one" hop represents connections at many levels of granularity, ranging from physical interactions in small protein complexes, to concepts as generic as "cell surface" (625 genes) or "GPCR signaling pathway" (960 genes). Thus, even though we only retrieve "1-hop" neighbors, the model has sufficient information to perform (effectively) multi-hop reasoning.
> > - As a direct result of high-connectivity nodes (large protein complexes, coarse annotations), the size of "multi-hop" neighborhoods increases exponentially. For example, if we only consider physical interactions from STRING and CORUM, the median "1-hop" neighborhood has 4 genes, while the median "2-hop" neighborhood has 4456 genes (relatively uninformative).
> >
> > **Baselines:**  Please see the general response. To summarize: As Table 1 demonstrates, directly applying NLP methods out-of-the-box on PerturbQA is comparable to random guessing. It is *non-trivial* to adapt these methods for molecular biology reasoning, especially because the vast majority of biological literature is behind paywalls (vs. in machine learning, where everything is open).
> >
> > **Information leakage:** Please see the general response. To summarize: Only ~3% of our testing pairs physically interact in any context, while only ~20% of pairs are directly connected by any annotation (including coarse ones). We find that the presence of physical interactions is minimally predictive of differential expression (DE).

---

> > > ### Comment · Reviewer_2VEF · 2024-11-26
> > >
> > > Thank you for your detailed response. While some of my concerns, such as multi-hop reasoning, have been addressed, significant issues remain unresolved. These include the lack of baselines, questions about the novelty of the proposed algorithm, the evaluation paradigm, and the limited experimental insights. Therefore, I will maintain my current score.

---

> > > > ### Author Response · Authors · 2024-11-26
> > > > **Thank you for your response**
> > > >
> > > > Thank you for your response and for reading our rebuttal!
> > > >
> > > > We would like to reiterate that the primary contribution of this work is not the proposed algorithm, but the framework for modeling perturbation experiments.
> > > >
> > > > As we write in the Introduction (Paragraph 2) and Background (Section 3), current approaches for modeling perturbations are misaligned with what biologists aim to glean from these large-scale screens.
> > > > We are the first to propose that predicting differentiation expression / direction of change and summarizing gene sets are more realistic endpoints, compared to existing regression-based objectives.
> > > >
> > > > Furthermore, our hypothesis was that language-based reasoning can be helpful for perturbation modeling. To demonstrate this, we developed a **lightweight, proof of concept**, that was evaluated against the **state-of-the-art in perturbation modeling**. These include graph (GEARS, GAT), language (GenePT), and pretrained single-cell (scGPT, in rebuttal) baselines. We are the first to integrate **experimental data alongside textual knowledge graphs** towards this application.
> > > >
> > > > There are many approaches in modern NLP that could be adapted towards perturbation modeling, and **this work exists to encourage and facilitate such exploration**, but we view this as beyond the scope of the current paper. We have prepared and presented perturbation screens / relevant knowledge sources in approachable formats for this purpose.
> > > >
> > > > Finally, we have provided additional insights, analyses, and human evaluation in our revised manuscript, so it would be very helpful to understand what concerns you feel remain unaddressed. Thank you for your time!

---

### Official Review · Reviewer_JGsH · 2024-11-03

**Soundness:** 3
**Presentation:** 3
**Contribution:** 2
**Rating:** 6
**Confidence:** 3

**Summary:**

The paper introduces a benchmark for evaluating LLM reasoning on biological perturbation experiments, and an LLM-based framework that uses knowledge graphs and prior data to outperform current methods in interpretability and performance on these tasks.

**Strengths:**

1. The method appears to be sound, combining RAG and COT prompting with knowledge graphs for handling complex biological relationships.
2. The proposed ramework emphasizes interpretable outputs, which is beneficial to biological research.
3. The evaluation was thorough, with both graph and language-centric baselines across multiple datasets.

**Weaknesses:**

1. The model focuses on discrete perturbation outcomes and does not address combinatorial perturbations, which are common in biological studies.
2. The interpretability of the method comes at the cost of additional complexity in prompt engineering, and it is not known how the performance is sensitive to the prompt design.
3. The paper acknowledges limitations in current evaluation metrics for enrichment tasks. It would be better to see the utilization of new, domain-specific metrics.

**Questions:**

1. How well does SUMMER generalize to new or sparsely annotated biological datasets? Are there performance drops when applied to less-studied cell lines or organisms?
2. Did the authors perform an error analysis to determine whether specific types of perturbations or gene functions are harder for SUMMER to predict accurately?

---

> ### Author Response · Authors · 2024-11-20
> **Thank you for your review (1/2)**
>
> Thank you for your review and suggestions! We hope this response clarifies some of your questions, and we look forward to discussing with you.
>
> ## Combinatorial perturbations
>
> The lack of combinatorial perturbations is less a limitation of the method, but of evaluation.
>
> To craft a "proof of concept" for combinatorial perturbations, it would be relatively straightforward to summarize the neighborhoods of each participating gene, and prompt for any synergy/lack thereof.
>
> However, existing datasets for combinatorial perturbations are limited. The most commonly analyzed are Norman et al. [1] and Wessels et al. [2], each containing around 100 perturbation pairs. These two experiments operate in different modalities (CRISPR activation vs. inhibition), and since there are no alternatives for comparison, it is difficult to quantify the quality of these data. A major goal of this work was to craft a trustworthy benchmark for perturbation modeling, so we chose to focus on single perturbations instead. Therefore we consider combinations currently out of scope, but an opportunity for future work when better datasets are available.
>
> [1] Norman et al. Exploring genetic interaction manifolds constructed from rich single-cell phenotypes. Science. 2019.
>
> [2] Wessels et al. Efficient combinatorial targeting of RNA transcripts in single cells with Cas13 RNA Perturb-seq. Nat Methods. 2023.
>
> ## Domain specific evaluation
>
> Due to the open-ended nature of the gene set task, automated evaluation methods are limited in their ability to reflect practical utility. Since this paper focuses on providing value to biologists, we recruited a domain specialist (molecular biologist, trained in wet lab and computational biology) for this task (See C.1).
>
> Overall, the LLM-generated summary was equal or more informative than the classical gene set enrichment results in 92% of cases, and agrees with the independent annotator in 72% of cases.
>
> 1. In 21/25 cases, the biologist reported that the LLM-generated summary was more informative. In 2/25 cases, they contained the same amount of information; and in 2/25 cases, the gene set contained more information.
> 2. In 18/25 cases, the biologist reported that the LLM summary captured the same biology as the original human annotation (our ground truth labels).
>
> 1. In the 2 cases where the gene set contained more *information*, a list of specific protein complexes were discovered, e.g.
>     ```
>     EIF2AK4 (GCN2) binds tRNA, Aminoacyl-tRNA  binds to the ribosome at the A-site, 80S:Met-tRNAi:mRNA:SECISBP2:Sec-tRNA(Sec):EEFSEC:GTP is hydrolysed to 80S:Met-tRNAi:mRNA:SECISBP2:Sec and EEFSEC:GDP by EEFSEC, UPF1 binds an mRNP with a termination codon preceding an Exon Junction Complex, Translocation of ribosome by 3 bases in the 3' direction, Translation of ROBO3.2 mRNA initiates NMD.
>     ```
>     However, this information-rich output is hard to interpret, compared to the LLM output, which the annotator marked as agreeing with the label of "translation."
>     ```
>    Ribosomal Protein Components Involved in Translation: This gene set is comprised of components of the large and small ribosomal subunits, which are essential for protein synthesis and translation. These genes are involved in the assembly and function of the ribosome, facilitating the translation of messenger RNA into protein.
>    ```
>
> 2. In the 7/25 cases where the LLM summary differed from the human annotation, the LLM annotation tended to miss some high specific terms, e.g. "targets of nonsense-mediated decay" was generalized to "stress response," and "dysregulated lncRNA antisense transcripts" was generalized to "nuclear gene regulation." Related terms tend to be sparsely annotated in Gene Ontology, so this indicates that it would be useful to tune the granularity of generations in the future, or to generate multiple candidates for specific descriptions.

---

> > ### Author Response · Authors · 2024-11-20
> > **Thank you for your review (2/2)**
> >
> > ## Additional biological analysis
> >
> > Thank you for the recommendation! We have updated the draft with several additional analyses regarding the qualitative aspects of our framework (currently in Appendix C), including evaluation from a domain expert (above). Here is a summary.
> >
> > - Clusters that elude manual annotation tend to be smaller or exhibit lower agreement. On these, gene set over-representation analysis focuses on highly specific gene sets, which each cover subsets of these clusters, rather than the whole. The LLM takes the opposite approach, and its summaries tend to "lift" the description to higher levels of hierarchy (Table 8). The two strategies provide orthogonal information, though the LLM outputs may be more readable.
> >
> > - We analyzed 300 generations (3 trials of 100 DE examples) to understand common failure modes (detailed examples in C.3). Errors and inconsistencies primarily resulted from deductions backed by overly-generic information. For example, the LLM may list an excessively broad set of influences, e.g. "mitochondrial function, protein synthesis, or transcriptional regulation," which affect nearly everything.
> >
> >   In several instances, the LLM was also confused between concepts which may be loosely connected, but not in the same context. For example, Gene A is upstream of stress signaling, e.g. "oxidative stress," which is related to the mitochondria. However, Gene A is *not* responsible for healthy mitochondria function, and should not respond similarly to genes that are.
> >
> > ## Other questions
> >
> > **Sensitivity to prompt design:** Due to computational constraints, we were unable to run inference a large number of times to evaluate a diverse set of prompts, so the SUMMER prompts were not particularly tuned for performance. During model development, we did observe that the 8b model tends to have an upper limit on the effective prompt length, and excessively long prompts were harder to follow (this could be resolved with longer context models, presumably).
> >
> > **Less well-studied systems:** Of the cell lines we study, RPE1 is perhaps the least well-characterized (though it is not genuinely *rare*). RPE1 is a non-cancerous cell line, and the [Gene Expression Omnibus database](https://www.ncbi.nlm.nih.gov/gds/), there are 5.1% as many RPE1 datasets as K562. Performance on DE/Dir does not differ noticeably on RPE1 compared to other cell lines.

---

> > > ### Comment · Reviewer_JGsH · 2024-11-27
> > > **Response to Rebuttal**
> > >
> > > We thank the authors for their efforts in answering my questions. While you have addressed most of my concerns, the reliance on manual prompt design and the limited budget for prompt sensitivity evaluation prevent a higher remark on the method novelty and broadness of reseach influence. So I decide to maintain my score.

---

> > > > ### Author Response · Authors · 2024-11-27
> > > > **Thank you for your response and review!**
> > > >
> > > > Thank you for your time!

---

### Author Response · Authors · 2024-11-20
**To all reviewers**

Dear reviewers,

Thank you for reading our paper and providing thorough feedback. We have updated our paper with a number of illustrative examples, which we hope will provide further insight into the questions raised here.

We would also like to clarify the scope and contribution of this work, which we believe are central to its interpretation.

1. **The primary contribution of this work is PerturbQA, a carefully curated benchmark for language-based reasoning and scientific discovery, in the context of single-cell perturbations.** Compared to existing benchmarks, which focus on scientific coding tasks [1] or reasoning over known facts [2], PerturbQA is *predictive* in nature, and its tasks are unsolved. PerturbQA draws upon experimental assays and harmonized knowledge graphs to replicate the reasoning required to "connect the dots" between known biology and unanswered questions.

    **Information leakage is the primary argument against this vision. However, we believe that the experimental outcomes from which we derive PerturbQA are minimally represented in existing knowledge bases and LLM pretrained weights.** Biological knowledge graphs tend to report relationships that have been well-validated by targeted studies, rather than the outcomes of large-scale screens. The gene pairs whose relationship we query are very much a *superset* of genes with well-characterized relationships. Specifically, only ~3% of gene pairs in our test sets physically interact (row 2, in *any* context, including other animals), and only ~20% share any pathway/annotation, including at the coarsest levels (row 4).

    | | K562 | RPE1 | HepG2 | Jurkat |
    |-|-|-|-|-|
    | physical, DE | 0.094 | 0.063 | 0.075 | 0.106 |
    | physical, total | 0.032 | 0.025 | 0.027 | 0.029 |
    | network, DE | 0.214 | 0.204 | 0.218 | 0.253 |
    | network, total | 0.222 | 0.209 | 0.220 | 0.208 |

    There is little difference between the positive/negative pairs in terms of higher-level connectivity (row 3 vs. 4). Physically interacting genes are more likely to result in differential expression in our dataset (row 1 vs 2), but **having a physical interaction is minimally predictive of DE,** as AUC hovers around 0.5, while Ours is consistently better than random guessing (from Table 1).

    | | K562 | RPE1 | HepG2 | Jurkat |
    |-|-|-|-|-|
    | physical = 1 | 0.53 | 0.52 | 0.52 | 0.54 |
    | Ours | 0.60 | 0.58 | 0.61 | 0.58 |

    Finally, Nadig et al. 2024 was published strictly after we downloaded the knowledge graphs. While the cell lines in question have been studied in prior work, in other contexts, Nadig et al. 2024 released the *first* large-scale Perturb-seq screens in these two cell lines.

2. **Classic LLM reasoning strategies achieve near-random performance  out-of-the-box on PerturbQA, and it is non-trivial to adapt them in *domain-aware* ways.** Within the past two years, there has been a plethora of brilliant, inference-time LLM strategies, from in-context learning, to CoT, ToT, (Graph) RAG, and more. However, as we demonstrate for ICL and CoT, naively applying existing templates to biological reasoning leads to near-random performance. **This also demonstrates the the *lack* of answers within the pretrained weights themselves.** With regards to retrieval-based baselines, a vast amount of biological literature is unfortunately inaccessible behind paywalls, or otherwise [subject to terms unfavorable for AI development](https://www.genecards.org/). As a result, it is difficult to benchmark standard retrieval-based strategies on equal footing.

3. To demonstrate that language-based reasoning is *feasible* on PerturbQA, **SUMMER integrates standard LLM techniques with domain-specific ways to query structured knowledge.** To the best of our knowledge, **SUMMER is the first fully LLM-based method for unseen perturbation prediction and rationalization**, without relying on any external classifiers or embedding models. Throughout our work, we acknowledge that techniques like CoT and retrieval are common in current LLM systems. The key contribution lies in *what* information is useful to retrieve, and *how* to frame the prompts to encourage reasonable reasoning.

[1] Rein et al. GPQA: A Graduate-Level Google-Proof Q&A Benchmark. 2023.

[2] Laurent et al. LAB-Bench: Measuring Capabilities of Language Models for Biology Research. 2024.

---

### Meta-Review · Area_Chair_ViiQ · 2024-12-20

**Metareview:**

This is a new benchmark contributing to understanding the reasoning capability of large language models in structured biological data.
This task will be of interest to communities interested in the space of AI and Biology.
However the recommendation at this point is borderline
Since the contribution is not so much methodological, it is unclear to make a strong case for this paper. On the other hand, ICLR has  Datasets as one of the topics in the call for papers and one could fit this paper under that category.

**Additional Comments On Reviewer Discussion:**

In the rebuttal phase authors could address some of the concerns. One of the important points that was not satisfactorily answered is because of manual prompt design such benchmarks maynot see much utility.

---

### Decision · Program_Chairs · 2025-01-22

Accept (Poster)